# DiffPBR: Point-Based Rendering via Spatial-Aware Residual Diffusion

**Yiping Xie**[1], **Yuchi Huo**[2], **Yunlong Ran**[1], **Zijian Huang**[2], **Lincheng Li**[3], **Yingfeng Chen**[3],
**Jiming Chen**[1], **Qi Ye**[1,4]*
[1]Group of Networked Sensing and Control, Zhejiang University,
[2]State Key Lab of CAD&CG, Zhejiang University,
[3]Fuxi AI Lab, Netease, Inc., Hangzhou, China,
[4]Zhejiang Key Laboratory of Airspace Awareness and Autonomous Unmanned Systems

## Abstract

Neural radiance fields and 3D Gaussian splatting (3DGS) have significantly advanced 3D reconstruction and novel view synthesis (NVS). Yet, achieving high-fidelity and view-consistent renderings directly from point clouds—without costly per-scene optimization—remains a core challenge. In this work, we present Diff-PBR, a diffusion-based framework that synthesizes coherent, photorealistic renderings from diverse point cloud inputs. We demonstrate that diffusion models, when guided by viewpoint-projected noise explicitly constrained by scene geometry and visibility, naturally enforce geometric consistency across camera motion. To achieve this, we first introduce adaptive CoNo-Splatting, a technique for fast and faithful rasterization that ensures efficient and effective handling of point clouds. Secondly, we integrate residual learning into the neural re-rendering pipeline, which improves convergence, generalization, and visual quality across diverse rendering tasks. Extensive experiments show that our method outperforms existing baselines with an improvement of **3~5 dB** in rendered image quality, a reduction from **41 to 8** in GPU hours for training, and increase from **3.6 FPS to 10 FPS** (our one-step variant) in rendering speed.

## 1 Introduction

Rendering photorealistic images from colored point clouds is a longstanding challenge in computer graphics, with significant applications in virtual reality, cinematography, robotics, and autonomous driving. In traditional point-based rendering pipelines, 3D point clouds are first projected onto 2D image planes through camera transformations, followed by z-buffer rasterization (Ravi et al., 2020). While these techniques are efficient and well-integrated into conventional graphics workflows, they often suffer from artifacts like holes, aliasing, and surface discontinuities, which are induced by the sparsity of the point clouds.

To address these limitations, neural point-based graphics (neural PBG) approaches augment traditional rendering pipelines with learning-based components, significantly improving visual quality and robustness. These approaches, such as NPBG (Aliev et al., 2020) and RPBG (Zhu et al., 2024), follow a similar pipeline: starting with the triangulation of 3D points from multi-view 2D observations, where neural textures are initialized at each proxy point. These points are then rasterized onto the target view, followed by a CNN-based refinement module that reconstructs photorealistic images from the coarse renderings.

While these methods employ per-descriptor optimization using images captured from a single scene, they struggle to generalize across different scenes. To overcome this limitation, NPBG++ (Rakhimov et al., 2022) introduces an online aggregation process that updates the neural descriptors of each point based on input views, enhancing its cross-scene capabilities. Despite these advancements, rendering consecutive frames often results in flickering, artifacts, and discontinuous jumps. To mitigate view inconsistencies, several approaches integrate alternative 3D representations (e.g., 3D

---

*Corresponding author.

CNNs in NPCR (Dai et al., 2020) and 3D Gaussians in PFGS (Wang et al., 2024a)) to alleviate the errors caused by splatting discrete point clouds onto image planes.

Given the improvements brought by combining multiple representations, a fundamental question arises: is a point inherently a poor graphic primitive for rendering? With recent advancements in depth sensing technologies such as Time-of-Flight (TOF) cameras, stereo depth cameras, LiDAR, monocular depth estimation (Yang et al., 2024b), and 3D/4D reconstruction (Wang et al., 2024b; Zhang et al., 2024; Chen et al., 2025), point clouds have become a prevalent modality alongside RGB images. Given the increasing availability of point clouds, can we achieve a purely point-based rendering pipeline without the need for integrating other representations? At first glance, the discrete nature of point clouds makes rasterization prone to artifacts such as holes and aliasing. This highlights the fundamental challenge of a purely point-based pipeline: mitigating rasterization artifacts while maintaining multi-view consistency.

In image restoration tasks, diffusion-based approaches have recently demonstrated impressive performance (Shi et al., 2024; Jiang et al., 2024; Xia et al., 2023; Zhu et al., 2023), offering strong generalization capabilities and high-quality outputs. These attributes make them a promising foundation for developing a generalizable point-cloud renderer. However, directly applying image-based diffusion models to refine point-based renderings introduces several intrinsic challenges. First, the standard image restoration process typically operates per image, without considering viewpoint consistency. This leads to inconsistencies across different views of a scene, undermining the temporal and geometric coherence essential for multi-view rendering. These inconsistencies are particularly problematic when applied to point clouds, where the absence of explicit surface connectivity amplifies the sensitivity of rendering to viewpoint changes. Second, diffusion models in image restoration rely on the assumption of pure noise inputs, yet degraded renderings of point clouds often retain substantial structural and color information. Reconstructing these inputs from scratch is both unnecessary and computationally inefficient, as it requires recovering fine details that are already present in the scene. This inefficiency leads to excessive computational overhead, hindering the model's performance. Third, point clouds lack explicit surface connectivity, making point-cloud rendering an ill-posed problem that is highly sensitive to point-wise parameters such as scale. Poorly chosen parameters can lead to further artifacts and unreliable supervision signals, complicating the model's ability to generalize effectively.

To address these challenges, we propose DiffPBR, a generalizable and view-consistent point-based rendering framework composed of three complementary components. First, we replace the conventional i.i.d. Gaussian noise with view-aligned, geometry-aware noise maps generated via CoNo-Splatting. These structured noise maps encode geometric cues such as depth and occlusion, remaining consistent across viewpoints and enabling the diffusion model to produce spatially coherent outputs with minimal computational overhead. Second, we adopt a residual diffusion paradigm, where the model predicts a weighted combination of residual (the difference between the rendered and ground-truth images) and noise. This formulation focus on recovering missing details, improving inference efficiency and enhancing generalization across diverse rendering conditions. Third, we introduce an adaptive point-based renderer that globally adjusts point scales, ensuring faithful projection while fully exploiting the diffusion model's capacity for refinement.

To summarize, we provide the following contributions:

- We introduce a novel and compact neural rendering pipeline that utilizes points as the sole graphics primitives for rendering. It generates photorealistic and view-consistent images across a wide range of scenes.

- We propose a spatial-aware residual diffusion process to accelerate the training of lifting rasterized images to high-quality images and ensure consistent multi-view synthesis.

- We propose an adaptive splatting strategy that dynamically adjusts point scales, ensuring faithful splatting and fully harnessing the capacity of the diffusion model.

- Compared with the state-of-the-art, our method demonstrates an improvement of **3∼5 dB** in rendered image quality, a reduction from **41 to 8** in GPU hours for training, and an increase from **3.6 FPS to 10 FPS** in the rendering speed frequency.

## 2 RELATED WORKS

### 2.1 SCENE SPECIFIC NOVEL VIEW SYNTHESIS

Novel view synthesis based on multi-view images for one scene has long been a fundamental challenge in computer vision and computer graphics (Zhang & Chen, 2004). The evolution of this field benefits from continuous advancements in 3D scene representation. Voxel grids offer the advantage of representing arbitrary topological structures, and when combined with interpolation techniques, they can generate continuous representations (Upson & Keeler, 1988). Breakthroughs in neural volume reconstruction (Lombardi et al., 2019) have revitalized research interest in this direction. NPBG (Aliev et al., 2020) pioneered neural point-based rendering by first reconstructing point clouds and attaching feature descriptors to each point to encapsulate local geometric and appearance information. These points are then rasterized into target views, enhancing point rasterization results through learned neural textures. The RPBG (Zhu et al., 2024) method follows a similar pipeline but employs surface-consistent neural descriptors and a CNN refinement module to address temporal instability in point rendering. In contrast to explicit 3D representations using voxels or point clouds, recent advances in deep learning have driven significant progress in implicit representations (Xiang et al., 2021; Kellnhofer et al., 2021). NeRF (Mildenhall et al., 2021) achieves continuous mapping from spatial coordinates and viewing directions to radiance values through MLP-based implicit functions, employing volume rendering to synthesize novel views. 3D Gaussian Splatting (Kerbl et al., 2023) leverages differentiable ellipsoidal structures and modern CUDA ecosystems to achieve high-quality novel view synthesis in remarkably short timeframes. Despite the compelling quality of novel view synthesis with these graphic primitives, they are limited to scene-specific optimization. To achieve generalization to general scenes, these graphic primitives are incorporated into feedforward networks that are conditioned on input multiview images (Wiles et al., 2020; Wang et al., 2021; Yu et al., 2021a).

### 2.2 GENERAL POINT-BASED RENDERING

The use of points as fundamental rendering primitives, known as point-based graphics, dates back to the 1970s (Csuri et al., 1979; Levoy & Whitted, 1985). Points gained popularity due to their storage flexibility, typically carrying 3D coordinates, colors, and optionally normals or radii, etc. In the early 2000s, Pfister et al. (2000) employed surfel splatting techniques to blend overlapping regions, achieving high-quality and efficient point cloud rendering. However, traditional point cloud rendering methods exhibit severe artifacts and holes when processing sparse point clouds or handling inaccurate point predictions.

To address these limitations, researchers have integrated point cloud representation with deep learning approaches. NPBG++ (Rakhimov et al., 2022) introduces online descriptor aggregation that updates point features from input views. These aggregated descriptors are rasterized onto 2D image plane and decoded into novel view images using a neural network. TriVol (Hu et al., 2023), on the other hand, proposes to combine point graphic primitives with anisotropic voxel grids fusion and neural radiance fields for image rendering (Mildenhall et al., 2021). PFGS (Wang et al., 2024a) draws inspiration from 3DGS (Kerbl et al., 2023) by representing points as adaptive Gaussians and employing differentiable splatting to learn neural descriptors to improve view consistency. A shared concept of these works is acquiring 3D descriptors for each point primitive, either by direct aggregation of color features from pre-trained feature extractors or finetune/learning the descriptors by multiview images, where other representations may be incorporated to help the learning.

Despite the great progress made by these works, we have found that splatting the descriptors tend to show blurry results as features from multiview images itself are inconsistent and direct aggregation can cause blurry effects. Although finetune/learning these descriptors from multiview images may help reduce the issue, the rasterization of the descriptors in discrete space still faces the inherent issues induced by the sparsity of point clouds. Unlike this paradigm, we work purely on the original point space and its original color values. Our work introduces a diffusion-based framework with residual refinement of rasterized images, and we address the view consistency issue by injecting structural diffusion noise in 3D space whose attributes are adaptively adjusted with the point cloud density. This point-based rendering pipeline offers improved view consistency, convergence, and rendering quality without explicit temporal supervision.

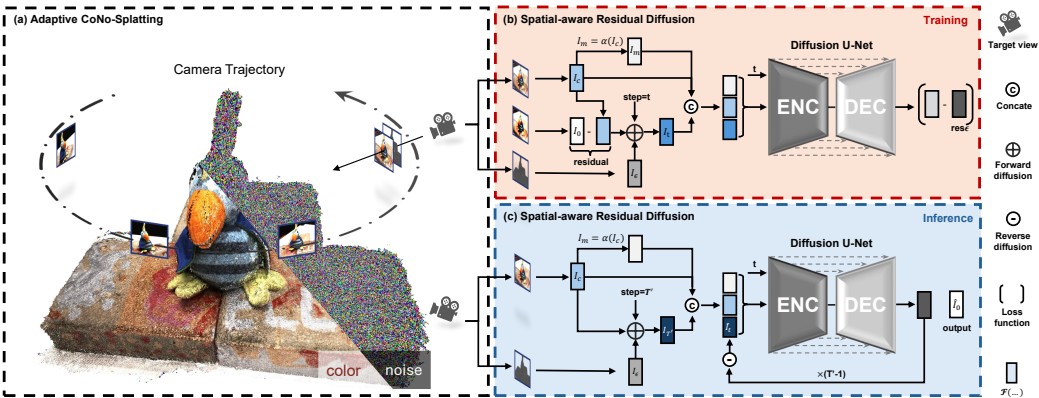

Figure 1: **Framework Overview**. Given a colored point cloud and calibrated cameras, DiffPBR synthesizes photo-realistic renderings via Adaptive CoNo-Splatting (Sec. 3.1) for view-consistent initialization, and a Spatial-aware Residual Diffusion stage (Sec. 3.2) for refinement.

# 3 METHOD

Given point cloud $\mathcal{P} = \{(\mathbf{x_i}, \mathbf{c_i}) \mid i = 1, 2, \ldots, n\}$, where $\mathbf{x_i} \in \mathbb{R}^3$ and $\mathbf{c_i} \in \mathbb{R}^3$ represent the 3D coordinates and color values of the points, we aim to synthesize photorealistic and view-consistent images from arbitrary camera perspectives.

While direct point-based rendering can produce plausible projections, the discrete nature of point clouds often leads to artifacts and inconsistent details across views. Inspired by recent advances in generative modeling (Shi et al., 2024; Jiang et al., 2024; Xia et al., 2023; Zhu et al., 2023), we leverage a diffusion model to remove rendering artifacts from point-based rasterization. However, standard diffusion models, such as Denoising Diffusion Probabilistic Models (DDPM) (Ho et al., 2020), transform Gaussian noise into clean data via iterative denoising. Despite their success in generation, they are often suboptimal for restoration tasks where the input already contains meaningful structure. In such cases, reconstructing the entire image from pure noise is inefficient and unnecessary. Moreover, the inherent stochasticity of the diffusion process often leads to poor temporal or multi-view consistency: since each sampling trajectory is initialized from random noise, predictions across adjacent viewpoints can diverge, causing noticeable inconsistency and temporal flickering.

To address these challenges, we harness 2D diffusion models for point-based rendering to improve both fidelity and cross-view consistency. Specifically, we propose a *spatial-aware residual diffusion module* (RDDM) with 3D-consistent noise guidance for coherent multi-view outputs. Based on RDDM, we further develop **DiffPBR**, a renderer consisting of two main components: (i) an adaptive point-based rasterizer that produces paired color renderings $I_c$, noise renderings $I_\epsilon$, and soft mask $I_m$, and (ii) a spatial-aware residual diffusion module $\mathcal{F}_\theta(\cdot)$ that enhances visual fidelity by recovering high-frequency details while enforcing cross-view consistency. An overview of the complete pipeline is provided in Fig. 1.

## 3.1 ADAPTIVE CONO-SPLATTING

In the first stage, we construct an adaptive point cloud $\mathcal{P}_{in} = \{(\mathbf{x_i}, \mathbf{c_i}, \epsilon_i, \mathbf{s_i}) \mid i = 1, 2, \ldots, n\}$ for appearance and noise rendering, respectively. Each point is initialized with a zero-mean Gaussian noise vector $\epsilon_i \in \mathbb{R}^3$ and an isotropic scale factor $\mathbf{s_i} = s_i \cdot (1, 1, 1)^\top \in \mathbb{R}^3$ that characterizes its contribution to the rendered pixels.

Let $\mathcal{K}_i$ denote the $k$-nearest neighbors of $\mathbf{x_i}$. The mean distance to $\mathcal{K}_i$ is first computed and used as the heuristic scale $\bar{s}_i$.

$$\bar{s}_i = \frac{1}{k} \sum_{\mathbf{x_j} \in \mathcal{K}_i} \|\mathbf{x_i} - \mathbf{x_j}\|_2, \quad s_i = \text{clamp\_max}\left(\bar{s}_i, \ \beta \cdot \text{median}\left(\{\bar{s}_j\}_{j=1}^N\right)\right), \quad (1)$$

where we adaptively clamp the local distance $\bar{s}_i$ to avoid excessively large values while preserving target-view consistency, using a global bound defined by the product of a learnable hyperparameter $\beta$ and the median of all local distances.

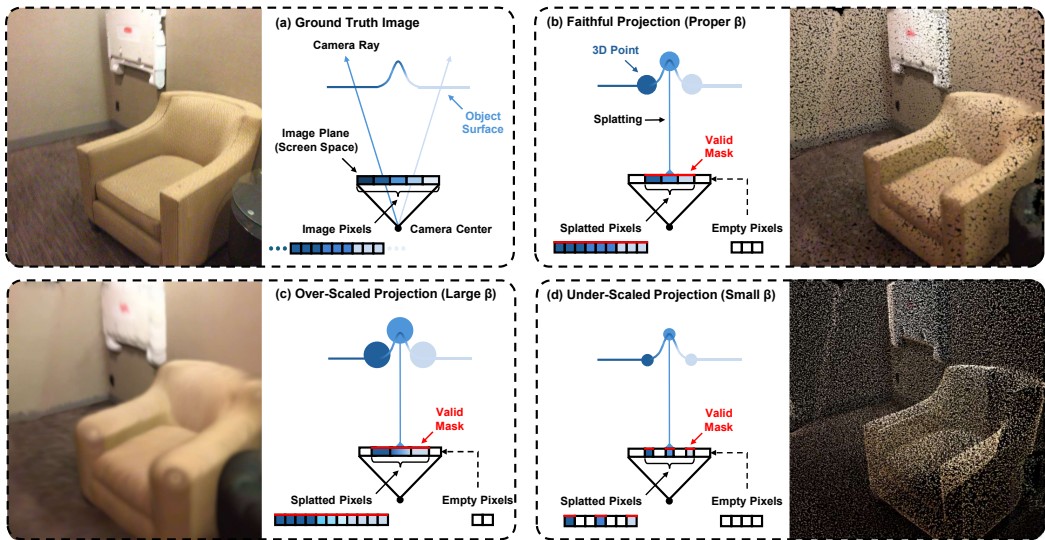

Figure 2: **Illustration of how point scale influences the distribution of valid pixels and holes in the splatted image.** (a) Ground-truth image with dense pixel coverage over the entire screen space. (b) With sparse points, the splatted image leaves empty regions, yet pixels within the valid mask remain well aligned with the ground truth. (c) Excessively large point scales cause splatting artifacts, thus penalizing $\mathcal{L}_{\text{cmp}}$. (d) Excessively small point scales result in screen holes and bleeding problems, thus penalizing $\mathcal{L}_{\text{cov}}$.

Balancing efficiency and informativeness, our design is simpler than previous point-based renderers. For example, PFGS requires a memory- and computation-intensive CNN-based module for per-point scale prediction, with a carefully crafted two-stage training strategy. This heavy design constrains their scalability and generalization ability. Conversely, our goal for scale initialization is not to directly generate photo-realistic renderings. Since the point cloud is inherently a sparse and discrete representation, missing regions cannot be recovered merely by enlarging splat sizes, which would instead introduce implausible artifacts. Rather, as depicted in Fig. 2 , our learnable global scale multiplier $\beta$ is designed to balance faithful information preservation with the avoidance of spurious artifacts during the splatting stage, ensuring that the splatted images provide effective input for the subsequent diffusion stage.

To achieve this goal, we first employ a differentiable point-based rasterization strategy that enables faithful projection of 3D points onto the image plane. We denote this rendering process as CoNo-Splatting, where each point, equipped with its color $\mathbf{c}_i$ and sampled noise $\epsilon_i$, is splatted onto the image plane as a circular footprint, producing the rendered RGB image $I_c^v$ and noise map $I_\epsilon^v$ at target view $v$:

$$I_c^v, \ I_\epsilon^v = \text{CNSplat}(\mathcal{P}_{in} \mid \mathbf{K}^v, \mathbf{M}^v). \tag{2}$$

Specifically, given the camera intrinsic matrix $\mathbf{K}^v$ and extrinsic matrix $\mathbf{M}^v$, the forward splatting step follows a general point-based rasterization formulation:

$$F(p) = \frac{\sum_{i=1}^n \kappa\Big((p - \pi(\mathbf{K}^v, \mathbf{M}^v, \mathbf{x}_i))/s_i\Big) v(z_i) \mathbf{f}_i}{\sum_{j=1}^n \kappa\Big((p - \pi(\mathbf{K}^v, \mathbf{M}^v, \mathbf{x}_j))/s_j\Big) v(z_j) + \delta}, \qquad \forall p \in \Omega. \tag{3}$$

Here, $p$ denotes a pixel location on the feature plane $\Omega \in \mathbb{R}^6$; $\pi(\cdot)$ is the standard pinhole projection mapping a 3D point to its 2D location; $\kappa(\cdot)$ is a differentiable splatting kernel distributing each point's contribution to nearby pixels; $z_i$ is the depth of point $i$ in the camera coordinate frame; $v(\cdot)$ is a generic differentiable visibility weighting function; and $\delta > 0$ ensures numerical stability.

From the rendered feature map $F(p)$, the $I_c^v$ and $I_\epsilon^v$ are extracted as:

$$I_c^v(p) = [F(p)]_{1:3}, \qquad I_\epsilon^v(p) = [F(p)]_{4:6}, \qquad \forall p \in \Omega. \tag{4}$$

**Algorithm 1** Multi-step DiffPBR-Q Training

1: **Inputs:** views $V = \{(I_0^v, \mathbf{K}^v, \mathbf{M}^v)\}_{v=1}^m$, point cloud $\mathcal{P}_c = \{(\mathbf{x}_i, \mathbf{c}_i)\}_{i=1}^n$
2: **Hyperparameters:** training steps $K$, diffusion steps $T$, learning rate $\eta$
3: $P_e = \text{Initialization}(P_c)$
4: $T' = \arg\min_{i=1}^T \left| \sqrt{\overline{\alpha}_i} - \frac{1}{2} \right|$
5: **for** $k = 1, \ldots, K$ **do**
6:     **for** $v = 1, \ldots, m$ **do**
7:         sample $t \sim \text{U}(1, T')$
8:         $I_c^v, I_\epsilon^v = CNSplat(\mathcal{P}_{in} \mid \mathbf{K}^v, \mathbf{M}^v)$
9:         $I_m^v = \alpha(I_c^v)$, $\mathcal{L}_{cns} = \mathcal{L}(I_c^v, I_0^v, I_m^v)$
10:         $\hat{I}_t^v = \tau_t \cdot I_0^v + (1 - \tau_t) \cdot I_r^v + \rho_t \cdot I_\epsilon^v$
11:         $\text{res}\epsilon = I_\epsilon^v + \gamma_t / \beta_t \cdot I_r^v$
12:         $\mathcal{L}_{dm} = \|\text{res}\epsilon - \mathcal{F}_\theta(\hat{I}_t, I_c, I_m, t)\|_2$
13:         $\theta, \beta \xleftarrow{\eta} \nabla_{\theta,\beta}(\mathcal{L}_{cns} + \mathcal{L}_{dm})$
14:     **end for**
15: **end for**

**Algorithm 2** Multi-step DiffPBR-Q Inference

1: **Inputs:** cameras $C = \{(\mathbf{K}^v, \mathbf{M}^v)\}_{v=1}^m$, point cloud $\mathcal{P}_c = \{(\mathbf{x}_i, \mathbf{c}_i)\}_{i=1}^n$
2: **Hyperparameters:** diffusion steps $T$
3: $P_e = \text{Initialization}(P_c)$
4: $T' = \arg\min_{i=1}^T \left| \sqrt{\overline{\alpha}_i} - \frac{1}{2} \right|$
5: **for** $v = 1, \ldots, m$ **do**
6:     $I_c^v, I_\epsilon^v = CNSplat(\mathcal{P}_{in} \mid \mathbf{K}^v, \mathbf{M}^v)$
7:     $I_m^v = \alpha(I_c^v)$
8:     $\hat{I}_{T'}^v = c_{T'} \cdot I_c^v + d_{T'} \cdot I_\epsilon^v$
9:     **for** $t = T', \ldots, 1$ **do**
10:         $\hat{I}_{t-1}^v = \frac{1}{\tau_t} \cdot (\hat{I}_t^v - \frac{\beta_t}{\rho_t} \cdot \mathcal{F}_\theta(\hat{I}_t^v, I_c^v, I_m^v, t))$
11:     **end for**
12: **end for**
13: **Return:** renderings $\{\hat{I}_0^v\}_{v=1}^m$

*Note*: we simplify terms $\sqrt{\overline{\alpha}_t} = \tau_t$, $\sqrt{1 - \overline{\alpha}_t} = \rho_t$, $(1 - \sqrt{\overline{\alpha}_t})\rho_t = \gamma_t$, and $I_c^v - I_0^v = I_r^v$, respectively.

Next, we process the rendered results and generate a soft mask $I_m^v$ by marking empty image regions as invalid. Let $I_m = \alpha(I_c)$, where $\alpha(\cdot)$ extracts the per-pixel opacity from the rendered $I_c^v$. A discrete spatial distribution is then defined as:

$$p(i,j) = \frac{I_m^v(i,j)}{\sum_{i,j} I_m^v(i,j) + \delta}, \tag{5}$$

We then introduce two regularizers for optimizing $\beta$:

$$\mathcal{L}_{\text{cov}} = \mathbb{E}_{(i,j) \sim p}\left[\left\|I_c(i,j) - I_0(i,j)\right\|_1\right], \tag{6}$$

$$\mathcal{L}_{\text{cmp}} = \mathbb{E}_{(i,j) \sim p}\left[-\log\left(p(i,j) + \delta\right)\right]. \tag{7}$$

As illustrated in Fig. 2, $\mathcal{L}_{\text{cov}}$ (coverage loss) encourages larger spatial coverage by diluting reconstruction errors over the valid region, while $\mathcal{L}_{\text{cmp}}$ (compactness loss) discourages overly spread-out masks. Their weighted combination:

$$\mathcal{L}_{\text{cns}} = \lambda_{cov}\mathcal{L}_{\text{cov}} + \lambda_{cmp}\mathcal{L}_{\text{cmp}} \tag{8}$$

establishes a complementary interplay: $\mathcal{L}_{\text{cmp}}$ alone would drive $\beta$ toward an overly small mask, while $\mathcal{L}_{\text{cov}}$ counterbalances this tendency by promoting sufficient spatial extent. Together, the two terms jointly regulate $\beta$ toward a well-structured and stable solution.

### 3.2 SPATIAL-AWARE RESIDUAL DIFFUSION PROCESS

Once the rendering triplets $\{I_c, I_\epsilon, I_m\}$ are obtained, we integrate them into the training and inference stage of diffusion models. As described in Algorithm 1 and Algorithm 2, the color image $I_c$ and the mask image $I_m$ are utilized as conditioning inputs to the diffusion model, where $I_c$ preserves faithful appearance cues and $I_m$ enforces precise localization of the missing regions. In addition, the splatted noise map $I_\epsilon$, which embeds structural priors, is leveraged to construct the supervision target. A predictor $\mathcal{F}_\theta$ is then optimized by minimizing the discrepancy $\mathcal{L}_{\text{rdm}}$ between the ground-truth residual-noise $\text{res}\epsilon$ and the model prediction, formulated as:

$$\mathcal{L}_{\text{rdm}} = \mathbb{E}_{I_0, I_\epsilon, t}\left[\|\text{res}\epsilon - \mathcal{F}_\theta(\hat{I}_t, I_c, I_m, t)\|_2\right]. \tag{9}$$

This design brings several key advantages:

**(i) Residual learning improves generalization and efficiency.** Rather than reconstructing images from pure Gaussian noise, the residual diffusion model predicts the discrepancy between rendered and ground-truth images. This simplifies the learning objective, as the network only needs to recover missing high-frequency details or correct subtle distortions. Consequently, the learned representations generalize better across diverse scenes, while starting the reverse process from

Table 1: Quantitative evaluation of state-of-the-art point-based rendering methods on three benchmark datasets. † indicates our reproduction results of the method.

| Method | ScanNet | | | DTU | | | THuman2.0 | | |
|---|---|---|---|---|---|---|---|---|---|
| | PSNR↑ | SSIM↑ | LPIPS↓ | PSNR↑ | SSIM↑ | LPIPS↓ | PSNR↑ | SSIM↑ | LPIPS↓ |
| Pytorch3D | 13.62 | 0.528 | 0.779 | 12.15 | 0.525 | 0.682 | 20.26 | 0.905 | 0.337 |
| NPBG | 15.09 | 0.592 | 0.625 | 13.52 | 0.703 | 0.514 | 19.77 | 0.915 | 0.112 |
| NPBG++ | 16.81 | 0.671 | 0.585 | 22.32 | 0.833 | 0.327 | 26.81 | 0.952 | 0.062 |
| TriVol | 18.56 | 0.734 | 0.473 | 19.25 | 0.592 | 0.518 | 25.97 | 0.935 | 0.059 |
| PFGS | 19.86 | 0.758 | 0.452 | 25.44 | 0.901 | 0.164 | 34.74/35.88† | 0.983/0.985† | 0.009/0.006† |
| DiffPBR-E | 22.92 | 0.816 | 0.412 | 28.15 | 0.919 | 0.138 | 40.89 | 0.985 | 0.006 |
| DiffPBR-Q | **23.28** | **0.827** | **0.399** | **28.45** | **0.935** | **0.124** | **41.27** | **0.989** | **0.003** |

informative renderings—rather than random noise—substantially reduces the number of denoising steps, accelerating inference without quality loss.

**(ii) Structured noise maps encode geometric priors.** The noise rendered via CoNo-Splatting replaces i.i.d. Gaussian noise, implicitly embedding geometry: pixels receive stronger contributions from points closer to the camera, while distant or occluded points contribute less or are suppressed. Since the model predicts the residual noise, it must decode these geometric cues, thereby acquiring spatial awareness without explicit depth supervision.

**(iii) Noise consistency ensures coherent generation.** Because the rendered noise maps originate from the same 3D point cloud, they naturally preserve geometric consistency across viewpoints and time steps. During inference, this consistency serves as a stable guidance signal, maintaining coherence along camera trajectories. This is especially beneficial in video-based or multi-view synthesis, where standard diffusion models often suffer from frame-to-frame inconsistency due to stochastic noise initialization.

In summary, our design tightly couples point cloud rendering with diffusion training, allowing us to inject geometric priors directly into the generative process and improve both quality and consistency of the outputs. Please see Sec. A of the supplementary materials for further details of 3D consistent $I_\epsilon$

## 4 EXPERIMENT

### 4.1 IMPLEMENTATION DETAILS

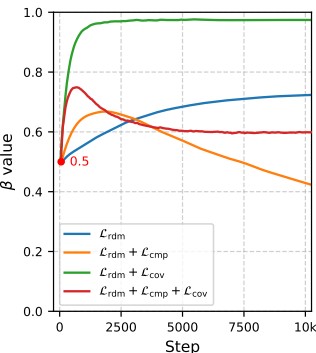

Figure 3: Adaptive $\beta$ value under different loss combinations.

**Baselines, Datasets & Metrics.** We employ a traditional graphics renderer Pytorch3D (Ravi et al., 2020), two point-based novel view synthesis methods NPBG (Aliev et al., 2020) and NPBG++ (Rakhimov et al., 2022), as well as a NeRF-based renderer TriVol (Hu et al., 2023) and a Gaussian-based renderer PFGS (Wang et al., 2024a) for comparison. Following PFGS, we evaluate DiffPBR on three datasets: a scene-level indoor dataset ScanNet (Dai et al., 2017), an object-level dataset DTU (Jensen et al., 2014), and a human body dataset THuman2.0 (Yu et al., 2021b), respectively. To assess reconstruction quality, we employ PSNR, SSIM (Wang et al., 2004), and LPIPS (Zhang et al., 2018).

**Training & Inference.** DiffPBR is trained in an end-to-end manner without stage-wise pretraining, using 8 NVIDIA GeForce RTX 3090 GPUs, while inference is performed on a single GPU. The training images are randomly cropped to $256 \times 256$, while the resolution is preserved during inference.

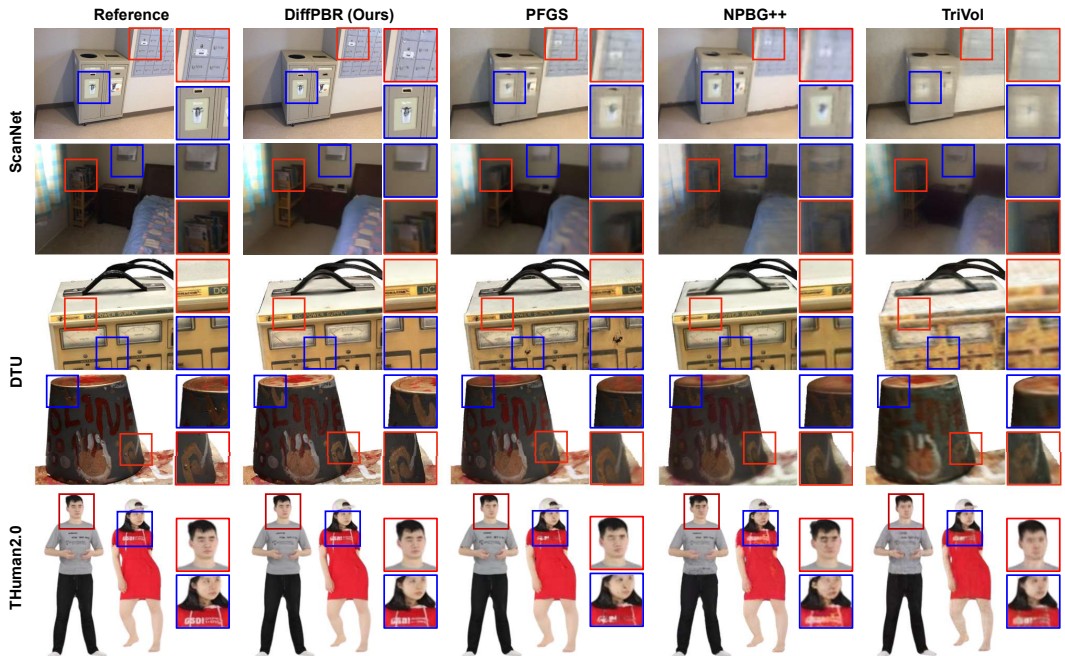

Figure 4: **Qualitative Results**. We show comparisons of ours to previous methods and the corresponding ground truth images from held-out test views.

**Model Variants.** Catering to different needs, we provide two versions of DiffPBR: **DiffPBR-Q** (Quality) with five sampling steps and **DiffPBR-E** (Efficiency) with a single-step sampling. A more detailed discussion is provided in Sec. B of the supplementary materials.

## 4.2 COMPARISON WITH BASELINES

**Evaluation of Rendering Quality.** To validate our method, we evaluate it against strong baselines on three challenging datasets. Quantitative comparisons are reported in Tab. 1, where the best and second-best results are highlighted in bold and underlined, respectively, while qualitative results are shown in Fig. 4.

Table 2: Evaluation of Model Efficiency on Thuman2.0.

| Method | Training | Inference | PSNR |
|---|---|---|---|
| PFGS | 41 | 3.6 | 35.88 |
| DiffPBR-E | $\sim 8$ | **10** | 40.89 |
| DiffPBR-Q | $\sim 8$ | 2 | **41.27** |

As shown in Tab. 1, our method consistently surpasses all competitors, with only marginal differences between the one-step and five-step variants. In Fig. 4, TriVol suffers from severe artifacts, while NPBG++ and PFGS better aggregate point features but at the cost of excessive smoothing that suppresses fine details. In contrast, our DiffPBR-Q combines point-based rendering with residual diffusion, yielding sharper and more realistic outputs. By implicitly modeling failure patterns in the rendered inputs and adaptively correcting them, it preserves pixel-level fidelity while enhancing generalization across datasets.

**Evaluation of Model Efficiency.** Tab. 2 reports rendering quality (PSNR) and efficiency—measured as training time (GPU hours) and inference speed (FPS)—for DiffPBR and PFGS on THuman2.0 with 80K points using an NVIDIA RTX 3090 GPU. Results show that both DiffPBR variants converge faster and yield higher rendering quality than PFGS, with the one-step version achieving about $3\times$ faster inference.

**Robustness with respect to Point Density.** We evaluate the robustness of our method under varying point densities on Thuman2.0. Training point clouds are randomly downsampled with ratios from 0.5

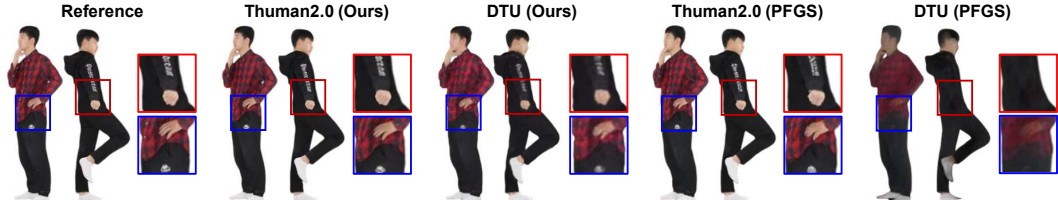

Figure 5: **Cross-dataset generalization**. Evaluation results labeled "DTU (X)" indicate that method X is trained on DTU without fine-tuning on Thuman2.0, whereas "Thuman2.0 (X)" refers to the in-domain setting.

to 1.0, while testing uses fixed ratios (1/2, 3/4, 5/6, 1.0). As shown in Tab. 3, our method consistently improves with higher input density and substantially outperforms PFGS.

Table 3: Robustness with respect to point density on Thuman2.0.

| Training | 60k $\sim$ 120k | | | |
|---|---|---|---|---|
| Inference | 60k | 80k | 100k | 120k |
| NPBG++ | 26.34 | 27.12 | 27.33 | 27.55 |
| TriVol | 25.93 | 26.21 | 26.71 | 27.08 |
| PFGS | 36.16 | 36.45 | 36.58 | 36.59 |
| DiffPBR-E | 39.87 | 40.96 | 41.29 | 41.72 |
| DiffPBR-Q | **40.02** | **41.29** | **41.62** | **42.01** |

**Evaluation of Cross-dataset Generalization.** To assess generalization, we directly evaluate our method and PFGS on Thuman2.0 using models trained on DTU. As shown in Fig. 5, our approach preserves finer details, whereas PFGS depends on a time-consuming pretraining stage to learn point cloud priors for Gaussian property prediction, which restricts its cross-dataset generalization. In contrast, by leveraging residual diffusion, our method can be trained on images with diverse degradations and achieves stronger generalization to unseen datasets.

## 4.3 Ablation Studies

Table 4: **Efficiency of Adaptive Splatting on Scan-Net.** We report both GPU runtimes and memory for processing 100k points.

| Method | Memory | Runtime |
|---|---|---|
| (a) KNN-based scale calc. | **0.094** | **5.35** |
| (a) + (b) adaptive scale reg. | 0.097 | 5.50 |
| (c) MLP-based scale pred. | 10.32 | 525 |
| (d) CNN-based scale pred. | 0.911 | 120 |

**Efficiency of Adaptive Splatting.** As described in Sec. 3.1, we initialize point scales with a heuristic KNN-based strategy, followed by an adaptive global scale regularization. For comparison, we implement two learnable alternatives based on—Multi-Layer Perceptron (MLP) and 3D CNN—for scale prediction. Tab. 4 presents the GPU runtimes and memory for processing 100k points, measured in milliseconds (ms) and gigabytes (GB), respectively. Our method directly computes scales without network inference, achieving a nearly $100\times$ speedup over MLP and $20\times$ over CNN, while requiring considerably less GPU memory.

Table 5: **Effect of Adaptive Splatting on ScanNet.** PSNR of rendered RGB images after CoNo-Splatting and diffusion refinement.

| Method | Splatting | Refinement |
|---|---|---|
| (a) KNN-based scale calc. | 13.43 | 22.05 |
| (a) + (b) adaptive scale reg. | 15.14 | **23.28** |
| (c) MLP-based scale pred. | 18.77 | 21.47 |
| (d) CNN-based scale pred. | 19.42 | 21.08 |

**Effect of Adaptive Splatting.** Following prior works, we pretrain both MLP- and CNN-based scale predictors. As shown in Tab. 5, predictors with priors yield splatted images closer to the ground truth (e.g., splatted images with CNN-predicted scales reach an average PSNR of 19.42). However, such strong inputs may become a double-edged sword for diffusion learning, contributing merely a 1.66 dB PSNR gain in refinement. In contrast, weaker initializations (e.g., KNN-based scales) encourage the model to learn richer geometry and textures under stronger gradients, while the dynamic scale constraint further promotes better parameter selection during optimization.

Table 6: Analysis of Adaptive Scale Regularizers

| | $w/o \, \mathcal{L}_{cns}$ | $w/o \, \mathcal{L}_{cov}$ | $w/o \, \mathcal{L}_{cmp}$ | Full |
|---|---|---|---|---|
| PSNR | 22.31 | 22.69 | 22.97 | **23.28** |

**Analysis of Adaptive Scale Regularizers.** As shown in Fig. 3 and Tab. 6, optimizing $\beta$ with diffusion loss alone is inefficient, yielding only a 0.26 dB improvement over direct KNN initialization (row 7). Incorporating

$\mathcal{L}_{cmp}$ degrades $\beta$ and, within a moderate range, introduces holes as shown in Fig. 2(b). This in turn drive the diffusion model to capture finer details. Conversely, $\mathcal{L}_{cov}$ encourages $\beta$ expansion and prevents the splatted images from collapsing to Fig. 2(d). Their adversarial balance is essential for achieving stable convergence and high-quality rendering.

Table 7: Effect of Modules in Residual Diffusion.

| Method | PSNR | Conv. |
|---|---|---|
| DDPM + 2D random noise | 19.22 | 104k |
| DDPM + 3D consistent noise | 20.07 | 92k |
| RDDM + 2D random noise | 21.54 | 44k |
| RDDM + 3D consistent noise | **22.15** | **37k** |

**Effect of Modules in Residual Diffusion.** As shown in Tab. 7, we compare diffusion strategies (DDPM vs. RDDM) and noise types (2D random vs. 3D consistent noise). RDDM not only improves PSNR by an average of 2 dB but also converges significantly faster, requiring only 5 sampling steps compared to 50+ for DDPM. Incorporating 3D-consistent noise further accelerates convergence while providing an additional 0.75 dB gain in PSNR by injecting structural priors from 3D space.

## 5 CONCLUSION

In this work, we present a diffusion-based framework for point cloud rendering that ensures temporal and spatial consistency via structured noise and residual denoising. By combining standard diffusion objectives with geometry-aware noise, our method naturally handles viewpoint variations without handcrafted temporal constraints. To our knowledge, this is the first application of residual diffusion in point cloud rendering, enhancing photorealism by refining noisy renderings toward realistic outputs.

## ACKNOWLEDGEMENTS

This work was supported in part by NSFC under No.62233013, 62293511, Key Research and Development Program of Zhejiang Province under No. 2025C01064, Fundamental and Interdisciplinary Disciplines Breakthrough Plan of the Ministry of Education of China under No. JYB2025XDXM103, CCF-NetEase ThunderFire Innovation Research Funding under No. 202305.

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

## APPENDIX

The contents of this appendix include:

## A  FORMAL DEFINITION AND FURTHER ELABORATION OF SYMBOLS

Tab. 8 provides formal definitions that ground the main paper and Appendix. Below, we further detail several key concepts.

**Noise attribute** $\epsilon_i$. For each point, we introduce a 3D noise vector $\epsilon_i$, sampled i.i.d. from a Gaussian distribution rather than inferred from spatial neighbors. This vector is concatenated with the point's color attributes and splatted once into the target view, yielding a six-channel feature map composed of RGB and noise. The noise vectors are initialized a single time before rendering and remain fixed thereafter. As shown in Fig. 6, the resulting noise maps exhibit consistent variations across viewpoints while implicitly encoding geometric cues such as depth and occlusion, thereby providing stable and informative guidance for the diffusion training and inference.

**Global scale regularizer** $\beta$. After a KNN-based heuristic initialization of per-point scales, we introduce a learnable scale regularizer $\beta$ that truncates point scales exceeding a global upper bound. Instead of per-point scale regularization—which incurs substantial training and inference overhead and scales poorly with larger point clouds—our single global parameter provides an efficient and adaptive mechanism for scale control. By jointly optimizing $\beta$ within the diffusion training process, the model learns to automatically calibrates point scales that coherently align with the denoising objective, thereby eliminating the reliance on pretrained CNN/MLP priors and enabling more robust and adaptive training.

Table 8: Summary of the important symbol definitions.

| Symbol | Definition |
|---|---|
| $\mathbf{x}_i$ | position attribute of the $i$-th point |
| $\mathbf{c}_i$ | color attribute of the $i$-th point |
| $\epsilon_i$ | noise attribute of the $i$-th point |
| $\mathcal{K}_i$ | $k$-nearest neighbors of the $i$-th point |
| $\bar{s}_i$ | heuristic scalar scale of the $i$-th point calculated by KNN |
| $\beta$ | learnable global scale regularizer |
| $s_i$ | adaptive scalar scale of the $i$-th point truncated by $\beta$ |
| $\mathbf{s}_i$ | isotropic scale vector of the $i$-th point |
| $\mathcal{P}$ | point cloud with only position and color |
| $\mathcal{P}_{in}$ | point cloud extended with adaptive scale $\mathbf{s}_i$ and 3D noise $\epsilon_i \overset{\text{i.i.d.}}{\sim} \mathcal{N}(0,1)$ |
| $v$ | index of camera perspective $v \in [1, m]$ |
| $t$ | index of diffusion step $t \in [0, T']$ |
| $T$ | total diffusion steps |
| $T'$ | truncated diffusion steps |
| $\mathbf{K}^v$ | $3 \times 3$ intrinsic matrix of camera view $v$ |
| $\mathbf{M}^v$ | $4 \times 4$ extrinsic matrix of camera view $v$ |
| $I_\epsilon^v$ | rendered consistent noise at view $v$ |
| $I_c^v$ | rendered RGB image at view $v$ |
| $I_0^v$ | ground-truth RGB image at view $v$ |
| $I_r^v$ | residual image defined as $I_c^v - I_0^v$ |
| $I_m^v$ | soft mask of $I_c^v$ at view $v$ |
| $\hat{I}_t^v$ | intermediate RGB image at view $v$ and step t |
| $\hat{I}_0^v$ | output refined RGB image at view $v$ |
| $\beta_t$ | variance schedule at step $t$ |
| $\alpha_t$ | signal preservation coefficient at step $t$ |
| $\bar{\alpha}_t$ | cumulative product of $\alpha_t$ up to step $t$ |
| $\tau_t$ | compact notation defined as $\sqrt{1 - \bar{\alpha}_t}$ |
| $\rho_t$ | compact notation defined as $\sqrt{\bar{\alpha}_t}$ |
| $\gamma_t$ | compact notation defined as $(1 - \sqrt{\bar{\alpha}_t})\rho_t$ |
| $\mathcal{F}_\theta$ | diffusion U-Net backbone |
| $\mathcal{L}_{cov}$ | loss encouraging the expansion of $\beta$ |
| $\mathcal{L}_{cmp}$ | loss encouraging the contraction of $\beta$ |
| $\mathcal{L}_{cns}$ | weighted sum of $\mathcal{L}_{cov}$ and $\mathcal{L}_{cmp}$ |
| $\mathcal{L}_{dm}$ | loss measuring diffusion output accuracy |

**Algorithm 3** One-step DiffPBR-E Training

1 **Inputs:** views $V = \{(I_0^v, \mathbf{K}^v, \mathbf{M}^v)\}_{v=1}^m$, point cloud $\mathcal{P}_c = \{(\mathbf{x}_i, \mathbf{c}_i)\}_{i=1}^n$
2 **Hyperparameters:** training steps $K$, diffusion steps $T$, learning rate $\eta$
3 $P_e = \text{Initialization}(P_c)$
4 $T' = \arg\min_{i=1}^T \left| \sqrt{\bar{\alpha}_i} - \frac{1}{2} \right|$
5 **for** $k = 1, \dots, K$ **do**
6     **for** $v = 1, \dots, m$ **do**
7         $I_c^v, I_\epsilon^v = CNSplat(\mathcal{P}_{in} \mid \mathbf{K}^v, \mathbf{M}^v)$
8         $I_m^v = \alpha(I_c^v), \; \mathcal{L}_{cns} = \mathcal{L}(I_c^v, I_0^v, I_m^v)$
9         $\hat{I}_{T'}^v = \tau_{T'} \cdot I_0^v + (1 - \tau_{T'}) \cdot I_r^v + \rho_{T'} \cdot I_\epsilon^v$
10         $\hat{I}_0^v = \mathcal{F}_\theta(\hat{I}_{T'}^v, I_c^v, I_m^v, T')$
11         $\mathcal{L}_{dm} = \mathcal{L}_{rec}(\cdot) + \mathcal{L}_{lpips}(\cdot) + \mathcal{L}_{gram}(\cdot)$
12         $\theta, \beta \xleftarrow{\eta} \nabla_{\theta,\beta}(\mathcal{L}_{cns} + \mathcal{L}_{dm})$
13     **end for**
14 **end for**

**Algorithm 4** One-step DiffPBR-E Inference

1 **Inputs:** cameras $C = \{(\mathbf{K}^v, \mathbf{M}^v)\}_{v=1}^m$, point cloud $\mathcal{P}_c = \{(\mathbf{x}_i, \mathbf{c}_i)\}_{i=1}^n$
2 **Hyperparameters:** diffusion steps $T$
3 $P_e = \text{Initialization}(P_c)$
4 $T' = \arg\min_{i=1}^T \left| \sqrt{\bar{\alpha}_i} - \frac{1}{2} \right|$
5 **for** $v = 1, \dots, m$ **do**
6     $I_c^v, I_\epsilon^v = CNSplat(\mathcal{P}_{in} \mid \mathbf{K}^v, \mathbf{M}^v)$
7     $I_m^v = \alpha(I_c^v)$
8     $\hat{I}_{T'}^v = \tau_{T'} \cdot I_c^v + \rho_{T'} \cdot I_\epsilon^v$
9     $\hat{I}_0 = \mathcal{F}_\theta(\hat{I}_{T'}^v, I_c^v, I_m^v, T')$
10 **end for**
11 **Return:** renderings $\{\hat{I}_0^v\}_{v=1}^m$

*Note*: we simplify terms $\sqrt{\bar{\alpha}_t} = \tau_t$, $\sqrt{1 - \bar{\alpha}_t} = \rho_t$, $(1 - \sqrt{\bar{\alpha}_t})\rho_t = \gamma_t$, and $I_c^v - I_0^v = I_r^v$, respectively.

**Truncated Diffusion Steps** $T'$. Given total diffusion Steps T, the truncated time step $T'$ defined as:

$$T' = \arg\min_{i=1}^T \left| \sqrt{\bar{\alpha}_i} - \frac{1}{2} \right|. \tag{10}$$

Intuitively, the truncation strategy indicates that the reverse process of the residual diffusion model starts not from pure Gaussian noise (at step $T$), but from a semi-informative state that retains partial structure and semantics of the original image (at step $T'$), as illustrated in Fig. 7.

Mathematically, such truncation can be viewed as a smooth equivalence transformation of the diffusion process that retains its generative behavior while facilitating more efficient inference. Specifically, the redefined diffusion forward process can be formalized as:

$$I_t^v = \sqrt{\alpha_t} \cdot I_0^v + (1 - \sqrt{\alpha_t}) \cdot I_r^v + \sqrt{1 - \alpha_t} \cdot I_\epsilon^v, \tag{11}$$

which can be then reparameterized as:

$$I_t^v = \sqrt{\bar{\alpha}_t} \cdot I_0^v + (1 - \sqrt{\bar{\alpha}_t}) \cdot I_r^v + \sqrt{1 - \bar{\alpha}_t} \cdot I_\epsilon^v \tag{12}$$
$$= (2\sqrt{\bar{\alpha}_t} - 1) \cdot I_0^v + (1 - \sqrt{\bar{\alpha}_t}) \cdot I_c^v + \sqrt{1 - \bar{\alpha}_t} \cdot I_\epsilon^v. \tag{13}$$

Note that $I_0^v$ is unavailable during inference, making the direct computation of $I_T^v$ intractable. Fortunately, at the truncated step $T'$, the weighting coefficient for $I_0^v$ in the diffusion formulation can be approximated to zero (i.e., $2\sqrt{\bar{\alpha}_t} - 1 \approx 0$) given a predefined linear scheduler. This allows $I_{T'}^v$ to be estimated as:

$$I_{T'}^v \approx (1 - \sqrt{\bar{\alpha}_t}) \cdot I_c^v + \sqrt{1 - \bar{\alpha}_{T'}} \cdot I_\epsilon^v \tag{14}$$
$$\approx \sqrt{\bar{\alpha}_t} \cdot I_c^v + \sqrt{1 - \bar{\alpha}_{T'}} \cdot I_\epsilon^v. \tag{15}$$

Since $I_{T'}^v$ is explicitly computable, we restrict diffusion to steps $t \leq T'$, while truncating the remaining steps to eliminate potential risks. The reverse process thus starts from $T'$ and gradually fits the current estimate $I_t^v$ to the ground truth $I_0^v$, implicitly reducing the residual term $R$ between $I_c^v$ and $I_0^v$.

**3D consistent noise** $I_\epsilon^v$. Unlike standard 2D random noise, our 3D consistent $I_\epsilon^v$ is rendered from the 3D point cloud, and therefore inherits two critical properties: (i) **view consistency**, as noise patterns across different views remain mutually aligned due to their shared 3D origin; and (ii) **geometry embedding**, as the noise distribution is modulated by depth and point-scale variations, implicitly encoding structural cues.

Intuitively, the noise images rendered across multiple viewpoints in Fig. 6 demonstrate that $I_\epsilon^v$ preserves the coherence of cross-view and reveals the coarse geometry of the scene. To further assess the statistical properties of $I_\epsilon^v$, we compute the covariance of the produced noise, its cross-covariance

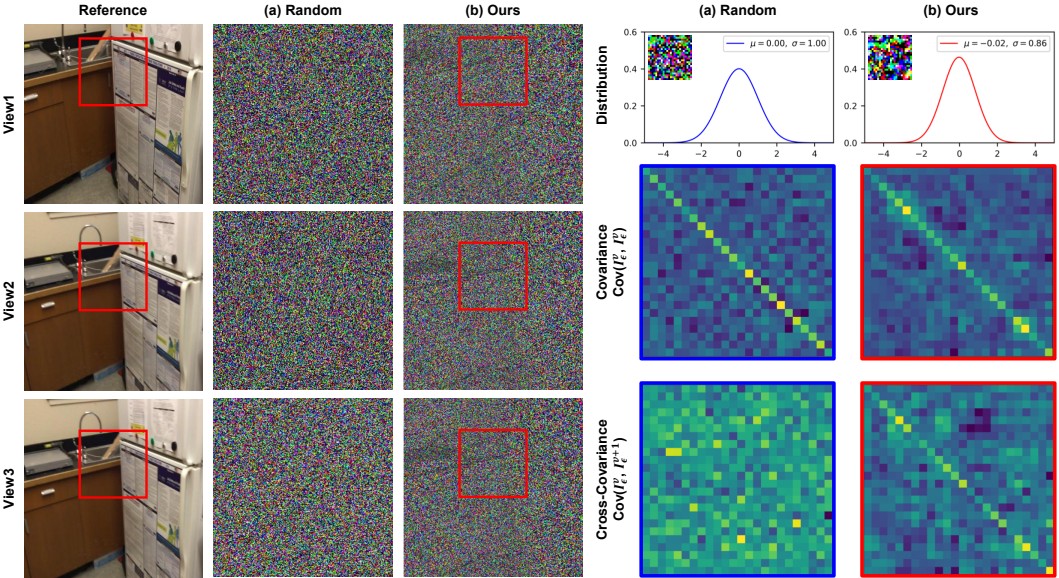

Figure 6: **Illustration of 3D consistent.** $I_\epsilon^v$ is rendered from 3D point cloud, ensuring multi-view consistency and geometric awareness while preserving i.i.d. Gaussian distribution properties.

with the noise of nearby viewpoint, and the distribution of the splatted noise values. The results show that random noise (a) does not show a correlation with nearby viewpoints, while our 3d consistent noise (b) preserves the Gaussian properties—including mean, variance, and i.i.d. nature—while faithfully representing the noise correlation between viewpoints, yielding an ideal 3d consistent noise map and remaining computationally efficient.

## B  DETAILS OF DIFFPBR VARIANTS

As mentioned in the main paper, we propose two complementary variants of DiffPBR, tailored to different trade-offs between quality and efficiency. Specifically, DiffPBR-Q adopts a multi-step sampling strategy to maximize rendering fidelity, while DiffPBR-E employs a single-step sampling scheme to prioritize efficiency.

As detailed in Sec. 3.2 of the main paper, the multi-step variant DiffPBR-Q adopts a residual diffusion pipeline, prioritizing reconstruction fidelity. To further enhance efficiency, we introduce DiffPBR-E, a one-step formulation. This variant is motivated by recent findings that diffusion models can remain effective with only a few iterations (Liu et al., 2024; Shi et al., 2024), and in some cases even a single step (He et al., 2024; Parmar et al., 2024). We provide the training and inference of DiffPBR-E in Algorithm 3 and Algorithm 4.

Rather than directly reducing the diffusion steps $T'$ to 1, we retain the full multi-step noise schedule but skip the intermediate steps during both training and inference. The resulting time step subset is defined as $\mathcal{T}_1 = \{1, T'\}$, a sub-sequence of the original schedule $\mathcal{T}_M = \{1, 2, \ldots, T'\}$.

Notably, the one-step version is trained to directly predict the ground-truth images (i.e., $x_0$-prediction), allowing the training to be supervised by more informative signals rather than merely regressing the residual noise.

As described in Algorithm 3, we supervise the training process with losses derived from readily available 2D supervision. The final loss includes the pixel-wise RGB L2 loss $\mathcal{L}_{rec}$ between the predicted and ground-truth image, given by:

$$\mathcal{L}_{rec} = \|I_0^v - I_\theta^v\|_2, \qquad (16)$$

the perceptual loss $\mathcal{L}_{lpips}$ based on the VGG-19 features (Simonyan & Zisserman, 2014), given by:

$$\mathcal{L}_{lpips} = \frac{1}{L} \sum_{l=1}^{L} \alpha_l \left\| \text{VGG}_l(I_0^v) - \text{VGG}_l(I_\theta^v) \right\|_1, \qquad (17)$$

and the Gram-matrix (GM) loss $\mathcal{L}_{gram}$ (Gatys et al., 2016) to enhance the sharpness of reconstructed images, given by:

$$\mathcal{L}_{\text{gram}} = \frac{1}{L} \sum_{l=1}^{L} \alpha_l \left\| \text{GM}_l(I_0^v) - \text{GM}_l(I_\theta^v) \right\|_1 , \tag{18}$$

where $\text{VGG}_l(\cdot) \in \mathbb{R}^{H \times W \times C}$ is the features from the $l$-th layer of a pre-trained VGG-19 network, $L$ is the number of layers considered, $\alpha_l$ is the weight of the $l$-th layer, and the Gram matrix at layer $l$ is defined as:

$$\text{GM}_l(I) = \text{VGG}_l(I)^T \text{VGG}_l(I). \tag{19}$$

During inference, the model operates in a non-iterative manner, yielding high-quality renderings without iterative sampling.

## C  IMPLEMENTATION DETAILS

### C.1  DATASETS

**ScanNet** is a large-scale indoor dataset comprising over 1,500 scenes and more than 2.5 million images, captured from different viewpoints and under varying quality conditions. We select the first 1,200 scenes for training and 300+ scenes for testing. During training, we aggregate the three nearest views to construct a view frustum point cloud from RGB-D images and their corresponding camera poses. This strategy decouples our method from the original scene scale, enabling efficient training on large-scale and complex scenes. At test time, we utilize the entire preprocessed sparse point cloud as input, ensuring consistency with the baseline methods.

**DTU** is a large-scale dataset for multi-view stereo. It includes over 100 scans taken under seven different lighting conditions and camera trajectories. We partition the dataset into 88 training scenes and 16 testing scenes, using an image resolution of $512 \times 640$. For each scan, we input the whole point cloud, which is constructed from the RGB-D images and subsequently down-sampled by a factor of 0.3 for sparsity.

**THuman2.0** dataset consists of 500 high-quality human scans captured using a dense DSLR rig. We use the first 75% of the scans for training and the remaining scans for testing, with an image resolution of $512 \times 512$. For each scan, we densely sample 1e6 points on gt mesh and use Pytorch3D to render 36 views as ground truth. During training and testing, the point clouds are down-sampled to 80k for their sparsity.

### C.2  MODEL ARCHITECTURE.

Our implementation of DiffPBR derives from Denoising Diffusion Probabilistic Model (Ho et al., 2020) in Pytorch, where the denoising module predicts pixel-level results. The core architecture is a U-Net-style encoder–decoder operating across four resolution scales. At each stage, time-conditioned ResNet blocks are combined with lightweight linear attention modules, enabling the model to efficiently capture both fine spatial details and temporal dynamics. A shared sinusoidal time embedding modulates all residual blocks, ensuring consistent and effective temporal conditioning throughout the network. The bottleneck incorporates a full-attention mechanism to aggregate global context, while skip connections facilitate the preservation of high-frequency structures across scales.

### C.3  TRAINING & INFERENCE.

To construct each training batch, we randomly sample camera views across all scenes to ensure diversity. For each selected view, we retrieve the corresponding point cloud required in the CoNo-Splatting process. To improve rendering efficiency, we perform view-dependent filtering by excluding 3D points that fall outside the target camera frustum, thereby reducing unnecessary computation while preserving relevant geometry. The rendered RGB images, noise maps, and corresponding ground truth are randomly cropped to a resolution of 256×256. To enhance the model's robustness to variations in lighting and viewpoint, we apply aggressive color augmentations—specifically, color jittering and grayscale conversion—as well as independent image-space flipping on each frame, even

within the same scene. The U-Net model is trained from scratch without incorporating any physical priors. The training loss is a weighted sum of the splatting and diffusion objectives:

$$\mathcal{L} = \lambda_{cov}\mathcal{L}_{cov} + \lambda_{cmp}\mathcal{L}_{cov} + \mathcal{L}_{dm}, \tag{20}$$

where $\lambda_{cov} = 0.01$ and $\lambda_{cmp} = 0.1$. The training process is performed on 8 NVIDIA RTX 3090 GPUs, requiring near 8 GPU hours.

At inference, we load the entire point cloud and initialize per-point scale and noise attributes once, thereby guaranteeing view-consistent color image and noise map across different camera views. To maintain compatibility with the U-Net architecture and prevent spatial misalignment, the rendered images are center-cropped such that both height and width are divisible by 16. For resolutions below 1K, a single NVIDIA RTX 3090 GPU supports efficient inference, making the model suitable for rapid prototyping and real-world applications.

## D    SYSTEM RUNTIME ANALYSIS

As mentioned in the main paper, DiffPBR begins with an adaptive CoNo-Splatting pipeline, followed by a diffusion refinement process. Consequently, the system's runtime primarily depends on three key factors: the number of points in the input point cloud, the resolution of the rendered image, and the number of function evaluations (NFE) during diffusion inference. We provide the detailed analysis in Fig. 7. All results are tested on a single NVIDIA RTX 3090 GPU.

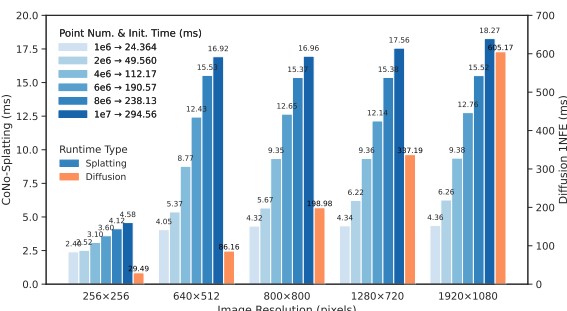

Figure 7: **Runtime evaluation of our system in different stages.** The blue bars indicate the runtime of the CoNo-Splatting process, with progressively darker shades of blue representing increasing point cloud density, while the orange bars denote the single-step forward inference time of residual diffusion (NFE=1).

**Splatting Stage.** The runtime of this stage consists of two components. First, we construct point cloud $\mathcal{P}_{in}$ with extended point attributes. In Fig. 7, we report the initialization time under different point densities. Subsequently, the generated feature points are rasterized onto the camera planes, producing a couple of images. The blue bars indicate the rasterization time under varying point densities and image resolutions. Except for the low-cost 256×256 resolution, the rasterization runtime remains relatively stable across different point densities at fixed resolutions. Conversely, when the resolution is fixed, the runtime increases slowly and approximately linearly with the number of points.

**Diffusion Stage.** The orange bars denote the single-step inference time of the U-Net at different image resolutions. The total runtime of this stage scales proportionally with the number of diffusion steps (NFE). Note that for resolutions of 1920×1080, diffusion inference on a single RTX 3090 encounters out-of-memory (OOM) issues. In this case, the single-step inference times are extrapolated based on trends observed in lower-resolution settings.

## E    ADDITIONAL EXPERIMENTS

### E.1    MORE ABLATION STUDIES.

**Diffusion Configurations.** As detailed in Appendix B, our multi-step diffusion model adopts $\epsilon$-prediction, whereas the one-step variant employs $x_0$-prediction. The diffusion configuration—including the parameterization type (i.e., prediction target) and the number of diffusion steps—is critical, as it not only defines the training objective but also impacts the inference procedure.

In this part, we conduct ablation studies on diffusion configurations by varying the prediction target ($\epsilon$ vs. $x_0$) and the sampling strategy (multi-step vs. single-step), as shown in Tab. 9. Across all training sizes, $\epsilon$-prediction with multi-step sampling consistently achieves the best PSNR, confirming its effectiveness in modeling fine-grained structures through progressive denoising. Within each

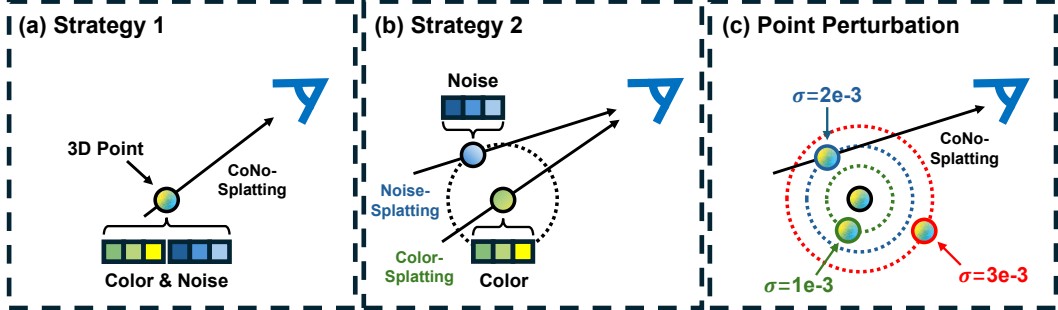

Figure 8: **Illustration of Point Cloud Initialization and Perturbation**. (a) The color of each point is concatenated with a randomly sampled noise vector and jointly splatted to the target view. (b) For each point, we additionally sample a new position and assign it a noise vector, after which the noise point cloud and the original color point cloud are splatted to the target view separately. (c) The clean point cloud is perturbed at different levels controlled by $\sigma$.

group, we observe that $\epsilon$-prediction benefits more from multi-step inference, while $x_0$-prediction performs slightly better with one-step inference. We attribute this difference to their respective training objectives: $\epsilon$-prediction is supervised purely with $\ell_2$ loss on the noise, which aligns well with iterative denoising; in contrast, $x_0$-prediction incorporates perceptual losses such as LPIPS and Gram loss in addition to pixel-wise $\ell_2$, encouraging high appearance fidelity that can be better preserved under direct (one-step) decoding. These observations highlight the roles of denoising strategy and prediction target in diffusion-based reconstruction.

Table 9: **Ablation studies on different combinations of diffusion configurations.** Each configuration is evaluated under different training set sizes on both THuman2.0 and ScanNet datasets.

| Config. | THuman2.0 | | ScanNet | |
|---|---|---|---|---|
| | Data | PSNR ↑ | Data | PSNR ↑ |
| $\epsilon$ + M. | 15k | **41.27** | 150k | **23.28** |
| $\epsilon$ + S. | 15k | 39.56 | 150k | 20.52 |
| $x_0$ + M. | 15k | 40.00 | 150k | 20.88 |
| $x_0$ + S. | 15k | 40.89 | 150k | 22.92 |
| $\epsilon$ + M. | 5k | **41.76** | 50k | **22.88** |
| $\epsilon$ + S. | 5k | 40.15 | 50k | 20.98 |
| $x_0$ + M. | 5k | 40.87 | 50k | 21.65 |
| $x_0$ + S. | 5k | 41.03 | 50k | 22.04 |
| $\epsilon$ + M. | 1k | **42.29** | 10k | **21.92** |
| $\epsilon$ + S. | 1k | 40.22 | 10k | 20.09 |
| $x_0$ + M. | 1k | 40.94 | 10k | 21.01 |
| $x_0$ + S. | 1k | 41.46 | 10k | 21.46 |

*Note*: we simplify the terms "$\epsilon$-prediction" as "$\epsilon$", "$x_0$-prediction" as "$x_0$", "Multi-step" as "M.", and "Single-step" as "S.", respectively.

Interestingly, we also observe that models trained on THuman2.0 achieve higher PSNR when trained on smaller subsets compared to larger ones. We attribute this to the simplified and spatially consistent degradation patterns in THuman2.0, which make it easier for the model to overfit and achieve better pixel-level reconstruction. While this leads to improved quantitative fidelity, it may come at the cost of generalization and robustness when applied to more diverse or complex scenarios.

To verify our hypothesis, we conduct additional experiments on the ScanNet dataset, which presents significantly higher complexity due to varying lighting conditions, motion blur, and incomplete as well as uneven point cloud sparsity. The results show that models trained on larger subsets consistently outperform those trained on smaller ones, highlighting the importance of data diversity for generalization.

**Point Cloud Initialization in CoNo-Splatting.** As described in the main paper, we augment each point's color with a zero-mean Gaussian noise vector, resulting in a noise point cloud $\mathcal{P}_\epsilon$ that retains the original geometry but carries a random appearance for subsequent noise rasterization.

Table 10: Initialization Strategy in CoNo-Splatting, with experiments conducting on Thuman2.0.

| Initialization | Noise Pos. | Noise Col. | PSNR ↑ |
|---|---|---|---|
| Strategy 1 | ✗ | ✓ | **41.27** |
| Strategy 2 | ✓ | ✓ | 40.02 |

In this part, we further explore the noise generation strategy where the position and color of each point are independently sampled from Gaussian distributions, aiming to assess how different noise patterns impact the training of the residual diffusion model.

As depicted in Fig. 8, in (a) we sample a three-channel Gaussian noise vector and concatenate it with the point color. The combined feature is then

Table 11: **CoNo-Splatting Configurations.** Evaluation metrics for different choices of $\lambda_{cov}$ and $\lambda_{cmp}$.

| $\lambda_{cmp}$ | 0 | 0.01 | 0.1 | 1 | $\lambda_{cov}$ | 0 | 0.01 | 0.1 | 1 |
|---|---|---|---|---|---|---|---|---|---|
| PSNR↑ | 22.31 | 22.57 | **22.69** | 22.39 | PSNR↑ | 22.31 | **23.01** | 22.97 | 22.62 |
| SSIM↑ | 0.804 | 0.815 | **0.821** | 0.806 | SSIM↑ | 0.804 | **0.826** | 0.824 | 0.810 |
| LPIPS↓ | 0.424 | 0.412 | **0.406** | 0.417 | LPIPS↓ | 0.424 | **0.402** | 0.409 | 0.419 |

splatted to the target view as a whole. In (b), each point is instead modeled as a local Gaussian distribution in 3D, with variance defined as the mean distance to its $k$ nearest neighbors. To render point noise, a new position is sampled from this distribution to replace the original point, whereas the position for point color remains fixed for color splatting.

As shown in Fig. 9 and Tab. 10, strategy 1 consistently outperforms strategy 2. The strict alignment of color and noise achieved by the joint splatting process in (a) leads to faithful preservation of fine structures. In contrast, the random positional noise introduced in (b) disrupts this alignment, resulting in degraded local detail reconstruction, as evidenced by blurred finger structures (highlighted in red boxes).

**CoNo-splatting Configurations.** As described in C.3 of the main paper, the training loss is a weighted sum of the splatting and diffusion objectives: $\mathcal{L} = \lambda_{cov}\mathcal{L}_{cov} + \lambda_{cmp}\mathcal{L}_{cmp} + \mathcal{L}_{dm}$, where $\lambda_{cov} = 0.01$ and $\lambda_{cmp} = 0.1$. Here, we provide a more detailed ablation study on this hyper-parameter choice. Specifically, we set $\lambda_{cov}$ and $\lambda_{cmp}$ to zero individually to examine their influence. The results are summarized in Tab. 11. Note that $\mathcal{L}_{dm}$ is the primary objective of the diffusion model, while these two regularizers are complementary. We assign them relatively lower weights compared to $\mathcal{L}_{dm}$.

Table 12: **Ablation on the statistical metric used for global truncation.**

| Metric | Q1 (25%) | Q2 (50% Median) | Q3 (75%) | Mean |
|---|---|---|---|---|
| PSNR↑ | 22.41 | **23.28** | 22.91 | 23.19 |
| SSIM↑ | 0.803 | **0.827** | 0.819 | 0.822 |
| LPIPS↓ | 0.421 | **0.399** | 0.412 | 0.404 |

Moreover, we further ablate the choice of the statistical metric used for global truncation in Eq. 12. Specifically, we compare four options: the 25th percentile (Q1), the 50th percentile (median, Q2), the 75th percentile (Q3), and the Mean. As shown in Tab. 12, the median (Q2) achieves slightly better performance than the other statistics. Note that both the median and mean are widely adopted in practice; however, the median is inherently more robust to outliers and provides stronger suppression of extreme values, making it better aligned with our global truncation strategy.

**Effect of Spatial-Aware Residual Diffusion Module.** To further validate the efficacy of our Spatial-Aware Residual Diffusion module, we conducted an ablation study by replacing it with two common alternative image synthesis paradigms: Nearest-Neighbor (NN) interpolation and learning-based image inpainting. As shown in Fig. 10, the inherent sparsity and uneven distribution of input points leave holes in the initial rendered images, causing direct NN interpolation to produce blurry and indistinct results. We evaluated two learning-based inpainting strategies. Mask-guided models (e.g., SD-XL Inpainting (Podell et al., 2023), FLUX-ControlNet Inpainting (Alimama-Creative, 2024)) struggled to fill these regions, as their training focuses on small, dense missing areas, unlike the large, continuous holes in our inputs. Text-to-image models (e.g., GPT-4o (OpenAI, 2024)) can generate photorealistic outputs from text, but may deviate from the original image, violating multi-view consistency across viewpoints.

Table 13: **Effect of Adaptive CoNo-Splatting as a Plug-and-Play Module.**

| Method | PSNR↑ | SSIM↑ | LPIPS↓ | GPU hrs↓ | FPS↑ |
|---|---|---|---|---|---|
| PFGS | 19.86 | 0.758 | 0.452 | 5+36 | 3.6 |
| PFGS+A.C.N.S. | **21.05** | **0.782** | **0.433** | **2+17** | **4.5** |

**Note**: A.C.N.S. denotes Adaptive CoNo-Splatting.

**Effect of Adaptive CoNo-Splatting as a Plug-and-Play Module.** As described in the paper, our contributions are tightly coupled from both theoretical and empirical perspectives, but they can also be applied to other methods. Specifically, we integrate our Adaptive CoNo-Splatting module into the baseline PFGS. Originally, PFGS extracts point-wise features and predicts per-point parameters such as scale and opacity via a Gaussian attribute regressor; the predicted colors and features are then rasterized for refinement. In our experiments, we replace the Gaussian regressor and its rendering pipeline with our adaptive point-based splatting module. As shown in Tab. 13 and Fig. 11, this substitution significantly improves both

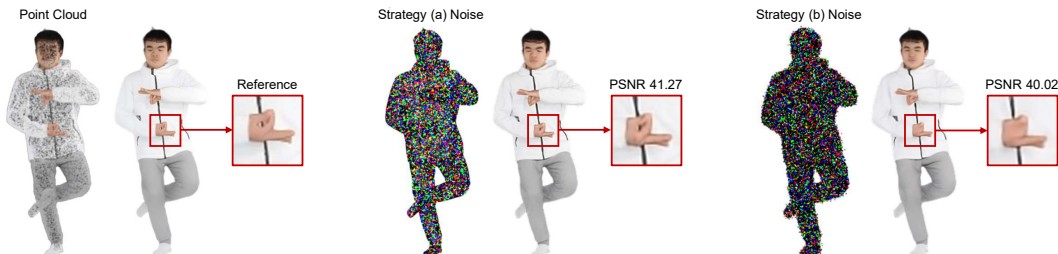

Figure 9: **Visualization of noise map and diffusion outputs**. The point clouds are initialized with strategies. Red boxes highlight degraded local details under strategy (b).

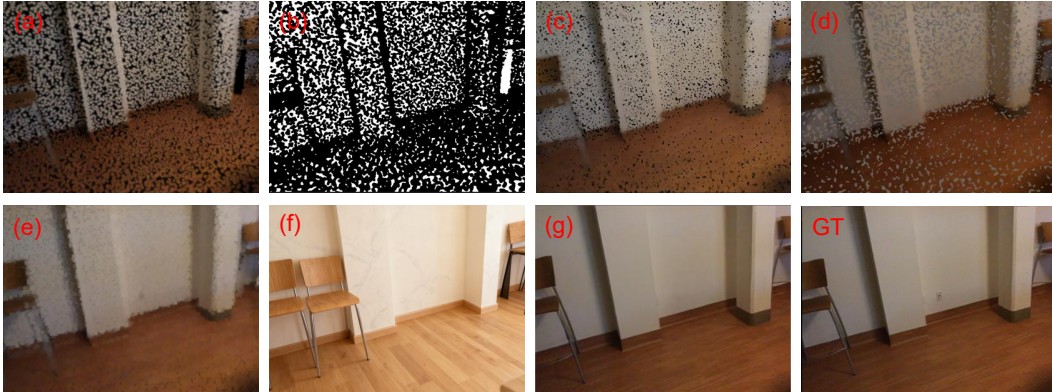

Figure 10: **Comparison of refinement module outputs.** (a, b) Image and mask from Adaptive CoNo-Splatting. (c, d) Inpainting results guided by (b) from FLUX-Controlnet-Inpainting and SD-XL-Inpainting, respectively. (e) Nearest-Neighbor (NN) interpolation on the image. (f) Output from GPT-4o. (g) Output from our method.

rendering quality and efficiency. Training and inference are faster thanks to the removal of convolutional layers in the Gaussian regressor, and the gap between the two training stages in the original PFGS is reduced, yielding better overall performance. These results demonstrate the versatility and effectiveness of our adaptive splatting design beyond its original context.

### E.2 MORE COMPARATIVE RESULTS.

Table 14: **Robustness with respect to varying point cloud perturbations on Thuman2.0.**

| Method | $\sigma = $ 1e-3 | $\sigma = $ 2e-3 | $\sigma = $ 3e-3 | Variance ↓ |
|---|---|---|---|---|
| PFGS | 33.80 | 33.09 | 32.14 | 1.89 |
| DiffPBR-E | 39.62 | 39.09 | 38.60 | 0.73 |
| DiffPBR-Q | **40.33** | **39.62** | **39.05** | **0.69** |

Table 15: **Evaluation of quality-efficiency trade-off on the DTU**. we report rendering quality, training time (GPU hours), and inference speed (FPS) at an image resolution of $640 \times 512$.

| Method | PSNR ↑ | SSIM ↑ | LPIPS ↓ | GPU hrs ↓ | FPS ↑ |
|---|---|---|---|---|---|
| NPBG++ | 22.32 | 0.833 | 0.327 | $\sim$10 | **11** |
| TriVol | 20.02 | 0.674 | 0.483 | $\sim$48 | 0.07 |
| PFGS | 25.44 | 0.901 | 0.164 | $\sim$40 | 0.5 |
| **DiffPBR-E** | 28.15 | 0.919 | 0.138 | $\sim$8 | 9.75 |
| **DiffPBR-Q** | 28.45 | **0.935** | **0.124** | $\sim$8 | 1.9 |

**Robustness with respect to Point Perturbations.** The results in Tab. 14 are evaluated under different perturbation levels, controlled by a variance factor $\sigma$ applied to the clean point cloud P. Ours outperforms PFGS across all settings, achieving higher PSNR with notably lower variance.

**System Runtime Comparison.** To evaluate the quality-efficiency trade-off, we compare the training time and inference speed (FPS) across methods on the DTU dataset at a resolution of $640 \times 512$. The competitors in our comparison generally follow a two-stage pipeline. In the first (fitting) stage, they extract information from the source imagee. For TriVol, this typically involves fitting the neural representation of the scene based on NeRF. For NPBG++, it corresponds to running a feature extractor on the selected neighboring views. Methods that rely on geometric proxies, such as PFGS, require the construction of a 3D representation as part of this stage.

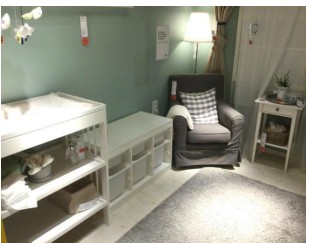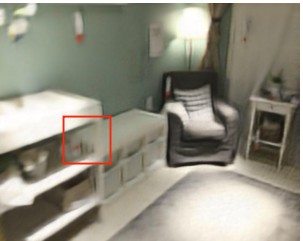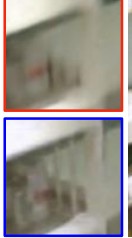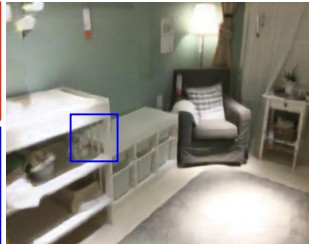

| Reference | PFGS | PFGS+Adaptive CoNo-Splatting |
|---|---|---|

Figure 11: **Effect of Adaptive CoNo-Splatting as a Plug-and-Play Module.** Incorporating Adaptive CoNo-Splatting leads to better preservation of local geometric structures, yielding more faithful reconstructions.

The time required for this process is included in our measurements to ensure a fair comparison. This fitting stage is performed only once per scene, after which the models can render arbitrary novel views—i.e., the second stage (rendering). We show a qualitative comparison between rendering quality and rendering speed (FPS) in Tab. 15. The results show that DiffPBR achieves the best rendering quality while maintaining a favorable trade-off in rendering efficiency.

**Rendering Quality Comparison with Generalizable Methods.** In Fig. 15 and Fig. 16, we further present qualitative results on THuman2.0, DTU and ScanNet. The results demonstrate that our method yields renderings that are more visually harmonious compared to competitors.

**Comparison with Per-scene Optimized Methods.** Tab. 19 and Fig. 13 present quantitative and qualitative comparisons between DiffPBR and state-of-the-art baselines, including two per-scene optimized approaches, 3DGS (Kerbl et al., 2023) and RPBG (Zhu et al., 2024). Notably, 3DGS regresses a set of Gaussian primitives, while RPBG learns neural point descriptors, both tailored separately for each training scene to achieve high-fidelity novel-view synthesis. In contrast, DiffPBR is trained over a diverse corpus of scenes and generalizes to previously unseen environments without scene-specific optimization. The results show that our method consistently outperforms both competitors on ScanNet, where the images contain motion blur and varying illumination. And on DTU, RPBG exhibits degraded performance in several scenes as shown in Fig. 13. On well-conditioned Thuman2 dataset, our performance is comparable to 3DGS and superior to RPBG. Note that due to the time-consuming per-scene training required by 3DGS and RPBG, the results reported in Tab. 19 are not evaluated on the full test set used in Tab. 1.

Table 16: **Evaluation on unbounded scenes on Tanks and Temples.**

| Method | Per-scene | PSNR↑ | SSIM↑ | LPIPS↓ |
|---|---|---|---|---|
| 3DGS | ✓ | 24.69 | 0.785 | 0.207 |
| RPBG | ✓ | 22.39 | 0.735 | 0.225 |
| PFGS | ✗ | 24.31 | 0.759 | 0.296 |
| Ours | ✗ | **25.25** | **0.801** | **0.157** |

**Evaluation on Unbounded scenes.** To further evaluate our performance on more diverse scenes, we conduct experiments on large scale unbounded scenes Tanks and Temples (Knapitsch et al., 2017). Specially, we compare with one of the generalizable method PFGS and two per-scene optimized method 3DGS and RPBG. We use the off-the-shelf feed-forward regressor Pi3 (Wang et al., 2025) to obtain the initial point clouds and corresponding camera poses from the input images. Tab. 16 and Fig. 14 present quantitative and qualitative comparisons. The results demonstrate that our method gets the best result compared to all other methods.

Table 17: **Bilateral comparison on dense and sparse point clouds from the Tanks and Temples Panther scene.**

| Method | Point cloud | PSNR↑ | SSIM↑ | LPIPS↓ |
|---|---|---|---|---|
| 3DGS | COLMAP | 22.16 | 0.726 | 0.300 |
| RPBG | COLMAP | 22.30 | 0.746 | 0.272 |
| Ours | COLMAP | **22.97** | **0.783** | **0.126** |
| 3DGS | Pi3 | 22.68 | 0.754 | 0.214 |
| RPBG | Pi3 | 22.93 | 0.780 | 0.181 |
| Ours | Pi3 | **23.49** | **0.854** | **0.109** |

Moreover, to further demonstrate the effectiveness of our approach under sparse point cloud inputs, we replace the dense point clouds produced by Pi3 with sparse COLMAP registration points. Specifically, we provide DiffPBR and the NVS baselines (3DGS and RPBG) with the same sparse-view COLMAP reconstructions. The results on the Panther scene are reported in Tab. 17.

**Comparison with Pure Graphics-based Renderer.** In the main paper, we compare our method with PyTorch3D, which performs differentiable rasterization-based rendering. To further broaden the comparison scope, we additionally include

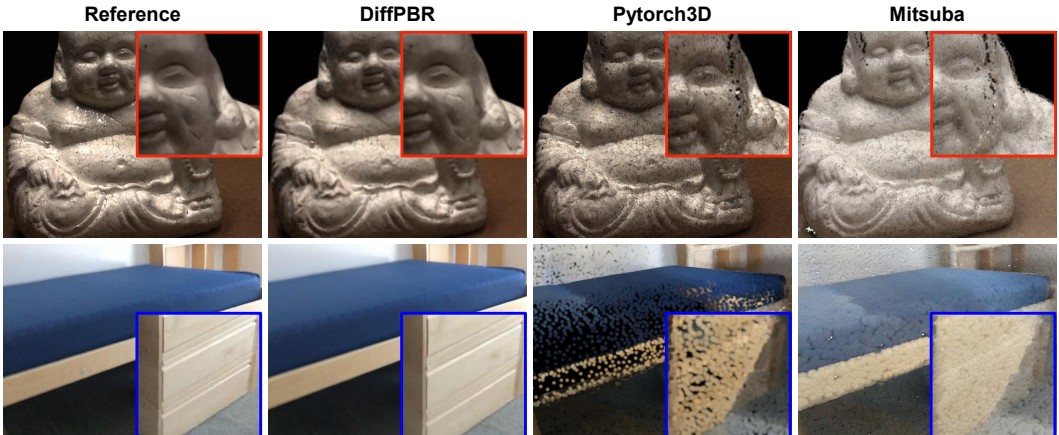

Figure 12: **Comparison with Pure Graphics-based Renderer on ScanNet and DTU**. Both graphics-based renderers fail to achieve photorealistic results due to point sparsity and structural noise.

Table 19: Quantitative evaluation of per-scene optimized point-based rendering methods on three benchmark datasets. Note that 3DGS and RPBG are per-scene optimized while ours is generalizable.

| Method | ScanNet | | | DTU | | | THuman2.0 | | |
|---|---|---|---|---|---|---|---|---|---|
| | PSNR↑ | SSIM↑ | LPIPS↓ | PSNR↑ | SSIM↑ | LPIPS↓ | PSNR↑ | SSIM↑ | LPIPS↓ |
| 3DGS | 31.29 | 0.896 | 0.219 | 24.92 | 0.871 | 0.191 | **41.57** | **0.989** | **0.004** |
| RPBG | 31.10 | 0.885 | 0.171 | 23.80 | 0.822 | 0.173 | 36.60 | 0.982 | 0.020 |
| Ours | **31.62** | **0.900** | **0.139** | **28.33** | **0.921** | **0.124** | 41.21 | 0.988 | **0.004** |

Mitsuba, a physically based ray-tracing renderer. Since Mitsuba simulates light transport through ray tracing, its illumination and shading are inherently inconsistent with rasterization-based pipelines such as PyTorch3D and ours. Therefore, we only provide qualitative comparisons for Mitsuba, as shown in Fig. 12, rather than quantitative metrics.

Table 18: **Multi-view consistency evaluation on ScanNet.** A higher TSED score indicates better multi-view consistency.

| T | Ours | PFGS | NPBG++ | TriVol | 3DGS | RPBG |
|---|---|---|---|---|---|---|
| 2 | **0.2347** | 0.2143 | 0.1837 | 0.1531 | **0.2347** | 0.2245 |
| 4 | **0.4796** | 0.4286 | 0.3776 | 0.3265 | 0.4694 | 0.4388 |
| 8 | **0.8776** | 0.7755 | 0.7143 | 0.6735 | 0.8571 | 0.8061 |

**Evaluation of Multi-view Consistency.** To assess the view-consistency of our model, we additionally report the Thresholded Symmetric Epipolar Distance (TSED) metric (Yu et al., 2023), which measures the proportion of frame pairs that satisfy epipolar consistency constraints. A smaller threshold T primarily captures temporal flickering or instability between adjacent frames, whereas a larger T reflects the global consistency of synthesized views under significant camera pose variations.

As shown in Tab. 18, our model consistently achieves higher TSED scores than all generalizable baselines and is comparable to or even better than per-scene optimized approaches, demonstrating superior multi-view consistency in novel view synthesis.

## F ADDITIONAL RELATED WORKS

In this section, we discuss additional related works on 3D consistent generation.

Directly applying 2D diffusion models view-by-view often yields view-dependent artifacts such as flicker, since independent noise trajectories produce uncorrelated stochasticity across viewpoints. To address this, recent work enforces 3D-consistent noise or consistency-aware denoising. One family of approaches exchanges information across views during denoising: ConsistNet introduces lightweight multi-view consistency blocks that unproject multi-view features into a global 3D volume and reproject consistent features back to each view to align parallel diffusion outputs (Yang et al., 2024a). Another class constructs structured or synchronized noise: ConsistDreamer generates 3D-

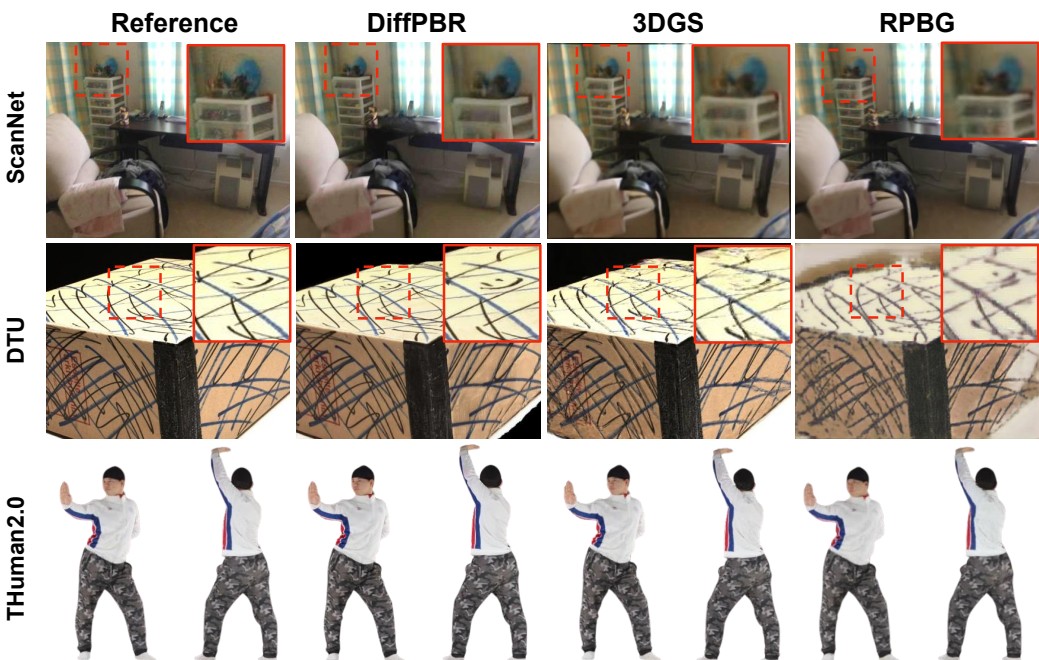

Figure 13: **Qualitative Results**. We show comparisons of ours to per-scene optimized methods and the corresponding ground truth images from held-out test views.

aware, structured noise and augments 2D diffusion with surrounding views and self-supervised consistency during training so that the stochastic component is correlated across views (Chen et al., 2024). Relatedly, Geometry-Aware Score Distillation (GSD) formulates a 3D-consistent noising process and a gradient-consistency loss in the score-distillation sampling (SDS) pipeline to reduce multiview gradient inconsistencies and improve geometry-aware text-to-3D optimization (Kwak et al., 2024). Complementarily, some methods design conditioning and training schemes on pretrained 2D priors to produce coherent multi-view images from a single input (e.g., Zero123++) (Shi et al., 2023). For temporal problems, EquiVDM demonstrates that using temporally consistent (warped) noise encourages equivariance and yields temporally coherent video diffusion outputs, suggesting an analogous benefit for multi-view/3D consistency when noise is correlated across frames or views (Liu & Vahdat, 2025). Finally, SyncNoise directly predicts geometry-consistent noise fields and leverages anchor-view propagation and depth supervision to further improve multi-view consistency for 3D editing (Li et al., 2024). While these strategies substantially reduce view-dependent artifacts, many trade off computational cost (per-scene finetuning or added modules). In contrast, our approach derives 3D-consistent noise guidance directly from the point cloud via adaptive CoNo-Splatting, enabling efficient residual 2D diffusion refinement without costly scene-level optimization.

# G  LIMITATIONS AND FUTURE DIRECTIONS

We present **DiffPBR**, a general residual diffusion framework for efficient and view-consistent refinement of point-based renderings with 3D-consistent noise guidance. A limitation of our method arises when a large portion of the point cloud is missing, since the model is currently trained from scratch and lacks strong image priors to hallucinate unseen regions. Another practical limitation is that the current implementation's rendering FPS is lower than real-time rates, primarily limited by the output resolution. Thanks to its flexible design, DiffPBR offers a scalable foundation that can benefit from larger model capacities, richer training datasets, and more diverse point-cloud sources such as LiDAR scans, depth projections, or mesh-based samples. We envision it as a step toward a generalizable and robust point-based renderer.

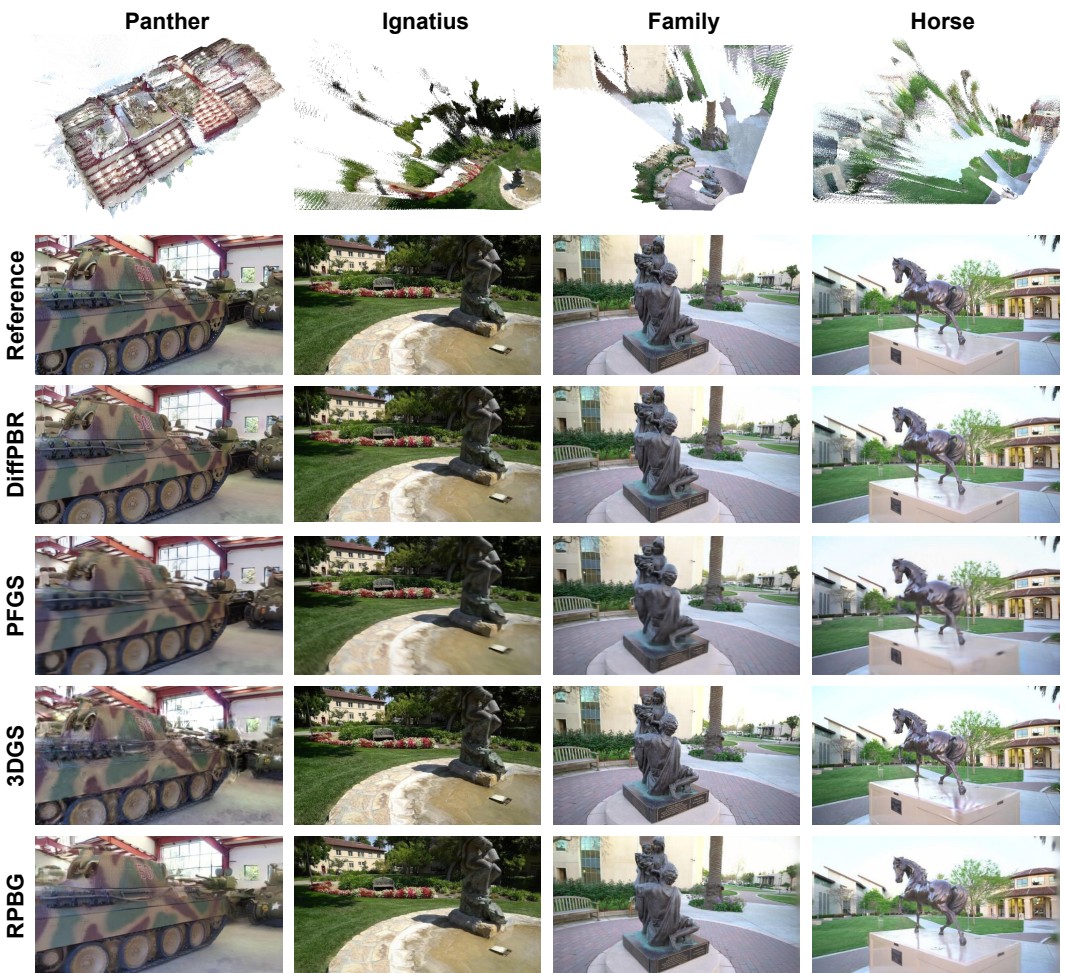

Figure 14: **Qualitative Results**. We show comparisons of ours to per-scene optimized methods and the corresponding ground truth images from held-out test views.

## H    FULL EVALUATION METRICS

We provide the full set of evaluation metrics here for the ablation studies and other experiments that previously reported only PSNR.

Table 20: Evaluation metrics for Tab. 3.

| Method | 60k | 80k | 100k | 120k |
|---|---|---|---|---|
| NPBG++ | 26.34/0.951/0.065 | 27.12/0.955/0.056 | 27.33/0.957/0.053 | 27.55/0.958/0.051 |
| TriVol | 25.93/0.933/0.059 | 26.21/0.935/0.054 | 26.71/0.938/0.052 | 27.08/0.939/0.049 |
| PFGS | 36.16/0.984/0.008 | 36.45/0.984/0.007 | 36.58/0.985/0.006 | 36.59/0.987/0.005 |
| DiffPBR-E | 39.87/0.983/0.007 | 40.96/0.985/0.005 | 41.29/0.987/0.004 | 41.72/0.988/0.003 |
| DIffPBR-Q | 40.02/0.985/0.006 | 41.29/0.989/0.004 | 41.62/0.990/0.003 | 42.01/0.990/0.003 |

Table 21: Evaluation metrics for Tab. 5, Tab. 6, and Tab. 7.

| Tab. 5 | | | Tab. 6 | | | Tabl. 7 | | |
|---|---|---|---|---|---|---|---|---|
| PSNR | SSIM | LPIPS | PSNR | SSIM | LPIPS | PSNR | SSIM | LPIPS |
| 13.43/22.05 | 0.519/0.805 | 0.782/0.422 | 22.31 | 0.809 | 0.418 | 19.22 | 0.742 | 0.468 |
| 15.14/23.28 | 0.602/0.827 | 0.633/0.399 | 22.69 | 0.815 | 0.414 | 20.07 | 0.783 | 0.447 |
| 18.77/21.47 | 0.735/0.796 | 0.470/0.428 | 22.97 | 0.820 | 0.404 | 21.54 | 0.796 | 0.425 |
| 19.42/21.08 | 0.740/0.782 | 0.466/0.433 | 23.28 | 0.827 | 0.399 | 22.15 | 0.808 | 0.420 |

Table 22: Evaluation metrics for Tab. 9.

| Thuman2.0 | | | ScanNet | | |
|---|---|---|---|---|---|
| PSNR | SSIM | LPIPS | PSNR | SSIM | LPIPS |
| 41.27 | 0.989 | 0.003 | 23.28 | 0.827 | 0.399 |
| 39.56 | 0.985 | 0.011 | 20.52 | 0.790 | 0.436 |
| 40.00 | 0.986 | 0.009 | 20.88 | 0.793 | 0.428 |
| 40.89 | 0.987 | 0.006 | 22.92 | 0.824 | 0.400 |
| 41.76 | 0.989 | 0.004 | 22.88 | 0.823 | 0.405 |
| 40.15 | 0.986 | 0.009 | 20.98 | 0.796 | 0.423 |
| 40.87 | 0.987 | 0.006 | 21.65 | 0.801 | 0.419 |
| 41.03 | 0.988 | 0.005 | 22.04 | 0.806 | 0.419 |
| 42.29 | 0.990 | 0.003 | 21.92 | 0.804 | 0.419 |
| 40.22 | 0.986 | 0.009 | 20.09 | 0.783 | 0.449 |
| 40.94 | 0.987 | 0.005 | 21.01 | 0.796 | 0.422 |
| 41.46 | 0.989 | 0.004 | 21.46 | 0.797 | 0.421 |

Table 23: Evaluation metrics for Tab. 10 and Tab. 14.

| Table 10 | | | Table 11 | | |
|---|---|---|---|---|---|
| PSNR | SSIM | LPIPS | PSNR | SSIM | LPIPS |
| 41.27 | 0.988 | 0.004 | 33.80 | 0.973 | 0.016 |
| 40.02 | 0.985 | 0.010 | 39.62 | 0.986 | 0.010 |
| | | | 40.33 | 0.987 | 0.007 |
| | | | 33.09 | 0.971 | 0.025 |
| | | | 39.09 | 0.985 | 0.012 |
| | | | 39.62 | 0.986 | 0.010 |
| | | | 32.14 | 0.967 | 0.031 |
| | | | 38.60 | 0.984 | 0.014 |
| | | | 39.05 | 0.985 | 0.012 |

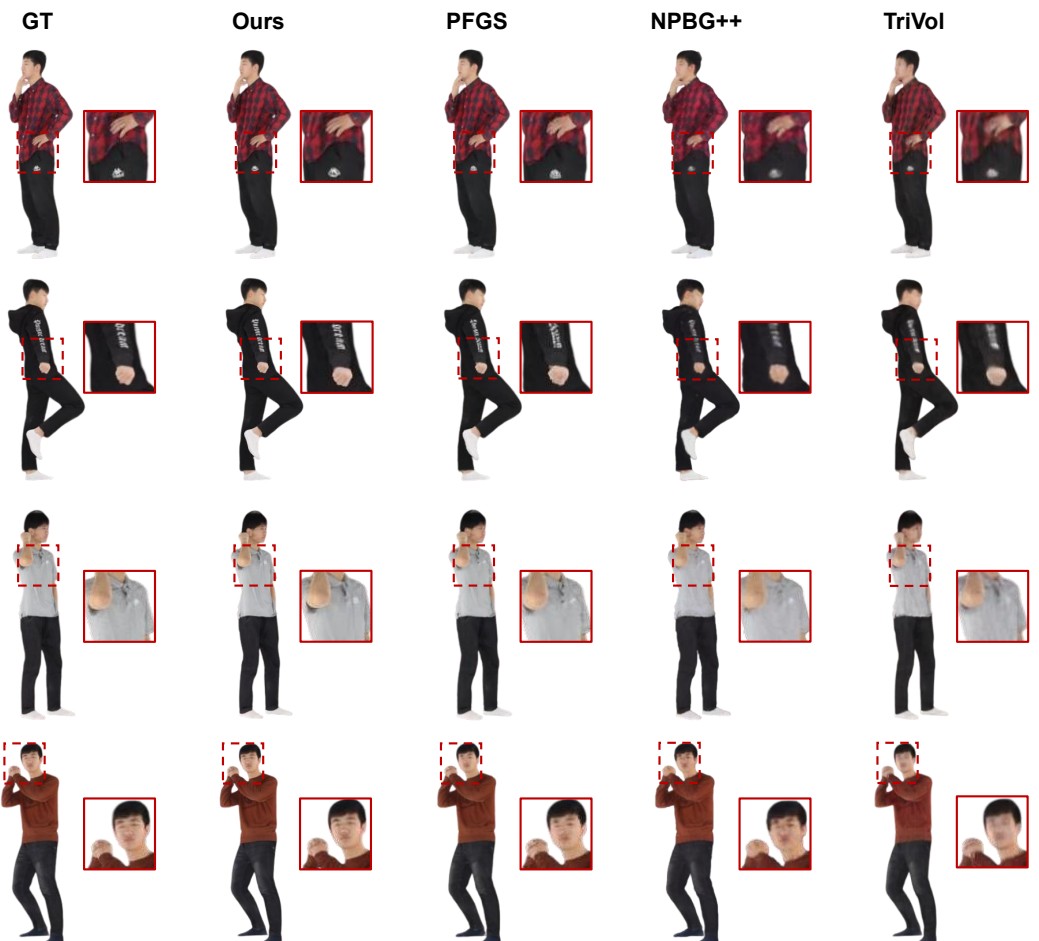

Figure 15: Additional qualitative comparisons between ours and baselines on Thuman2.0.

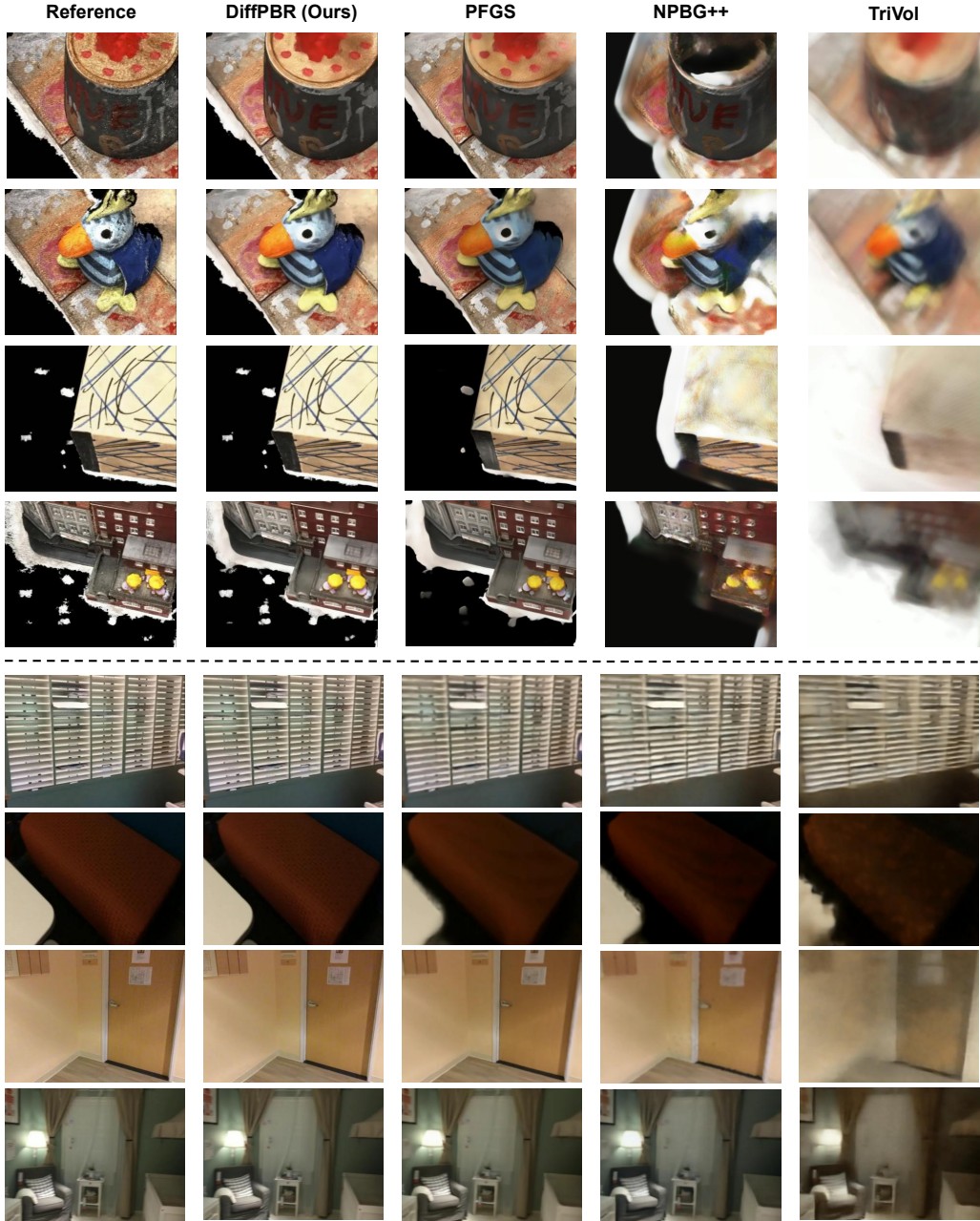

Figure 16: Additional qualitative comparisons between our method and baselines on DTU (upper block) and ScanNet (lower block).

