# OpenReview forum: "DiffPBR: Point-Based Rendering via Spatial-Aware Residual Diffusion"
_ICLR.cc/2026/Conference — ICLR 2026 Poster_

### Official Review · Reviewer_Vaw3 · 2025-10-29

**Soundness:** 3
**Presentation:** 3
**Contribution:** 3
**Rating:** 8
**Confidence:** 5

**Summary:**

This paper proposes DiffPBR, a diffusion-based framework for point-based rendering to produce photorealistic and view-consistent images directly from point clouds. The method introduces two core components: (a) Adaptive CoNo-Splatting, a differentiable point rasterizer that adaptively adjusts point scales using a learnable global regularizer; (b) Spatial-Aware Residual Diffusion (RDDM) – a diffusion model conditioned on geometry-aware, view-consistent noise maps and trained to predict residuals. The two model variants, -Q and -E, have shown large improvements over the baseline method of PFGS.

**Strengths:**

- The idea of applying noise mechanism in 3D space (rather than 2D) is very interesting to me, and the implementation does achieve satisfactory results on common benchmarks. The residual diffusion formulation is elegant and well-motivated, reducing computational cost while preserving fidelity.
- The pipeline is well-illustrated and the pseudo-code & the appendix demonstrate commendable clarity and reproducibility for readers.
- The proposed CoNo-Splatting has the potential to be developed as a common plug-in for most point-based rendering methods.

**Weaknesses:**

- The two claimed contributions—Adaptive CoNo-Splatting and Spatial-Aware Residual Diffusion—are somehow independent, with limited conceptual or algorithmic coupling. Each component could function as a standalone contribution, making the overall framework appear as a loose combination rather than a tightly integrated method.
- The evaluation of performance is not sufficient, in terms of both benchmarks and baselines. The three datasets are either bounded (ScanNet) or captured under well-controlled lab environment. More challenging cases such as Tanks and Temples are recommended to see the real potential of these methods. For baselines, I think the authors should at least include 3DGS as it's one of the most known method for point-based rendering.
- The rendering fps is not suitable for real-time applications, though it's not considered as a drawback of this work.

**Questions:**

- What does the PyTorch3D in the experiments particularly mean? Do the authors mean directly rasterizing points by PyTorch3D? If so, it shouldn't be a baseline method since PyTorch3D is just an implementation.
- What if Adaptive CoNo-Splatting is applied to other methods? I wonder its performance as a plug-and-play module.
- Also it seems that the authors have misused the tex commands of \citep{} and \citet{}. Please correct these citation formats in the modified manuscript accordingly.

---

> ### Author Response · Authors · 2025-11-28
> **Response to Reviewer Vaw3 (1/3)**
>
> We sincerely thank Reviewer Vaw3 for the time and effort in reviewing our work and providing supportive comment. We address each concern or question below.
>
> **Comment 1.** The coupling of the two claimed contributions—*Adaptive CoNo-Splatting* and *Spatial-Aware Residual Diffusion*—are somehow independent.
>
> **Response.** We thank the reviewer for raising this thoughtful concern. We kindly clarify that *Adaptive CoNo-Splatting* and *Spatial-Aware Residual Diffusion* are designed to be tightly coupled from the following perspectives:
>
> **1. Theoretical Coupling.** Our key insight is that **converting a deterministic sparse point cloud into a probabilistic representation provides stronger guidance for the refinement network**. Due to the inherent modality of point clouds—sparse and uneven—the direct point projection onto the image plane inevitably produces holes. Through *Adaptive CoNo-Splatting*, we transform this sparse representation into a relatively dense probabilistic distribution on the projection plane. **This probabilistic formulation enables the refinement network to learn both deterministic structures and uncertainty patterns, effectively guiding it to infer and fill the missing regions in the rendered image**. Specifically, the *Adaptive CoNo-Splatting* module dynamically adjusts per-point scales $\beta$ to provide *faithful* inputs—an RGB image as the conditioning input and a noise map as the forward input—for the subsequent *Residual Diffusion*. As illustrated in Figure 2(b), the missing regions in the images produced by *Adaptive CoNo-Splatting* naturally correspond to the residuals that the diffusion process is tasked with predicting.
>
> **2. Integration Benefits.** Since our splatted 3D-consistent noise originate from 3D point clouds, **they naturally encode geometry cues such as occlusion and depth relations** (see Appendix Figure 6). As the diffusion model predicts residual noise, it must decode these geometric cues, **thereby gaining *spatial awareness* without explicit depth supervision**. During inference, **the splatted 3D-consistent noise serves as a stable signal**, preventing mode collapse and maintaining coherence along camera trajectories.
>
> **3. Empirical Evidence.** As reported in Table 5, removing the dynamic scale constraint and using heuristic scale estimation degrades diffusion performance. Interestingly, when we replaced CoNo-Splatting with a pretrained CNN-based scale regressor (as in PFGS), the initial splatted PSNR improved (18.77 vs 15.00) but the final refined image PSNR dropped by 2 dB. This indicates that **weaker but faithful initializations encourage richer geometry and texture learning under stronger gradients, while the dynamic scale constraint promotes better parameter choice selection during optimization**. Additionally, Table 7 confirms that structured 3D-consistent noise significantly improves both the convergence and the quality of diffusion training compared to conventional 2D i.i.d. Gaussian noise.
>
> Overall, we believe that the two contributions—*Adaptive CoNo-Splatting* and *Spatial-Aware Residual Diffusion*—are **complementary and interdependent**, working together to form a unified rendering–refinement framework. We hope this clarification alleviates some of the reviewer’s concerns, and we are happy to provide further details if needed.
>
> **Comment 2.** More evaluation results in terms of both benchmarks such as *Tanks and Temples* and baselines such as *3DGS* would improve the completeness of the experimental analysis.
>
> **Response.** We thank the reviewer for this valuable suggestion. Following the recommendation, **we have included additional benchmarks (Tanks and Temples) and baselines (3DGS and RPBG [1])**, with detaied evaluation results and discussions included in Appendix E.2 of the revised manuscript.
>
> Results on Tanks and Temples
> | Method | Per-scene | PSNR ↑ | SSIM ↑ | LPIPS ↓ |
> |------:|:--------------:|-------:|-------:|--------:|
> | Ours  |×| **25.25** | **0.801** | **0.157** |
> | PFGS  |×| 24.31  | 0.759  | 0.296  |
> | 3DGS  |√| 24.69  | 0.785  | 0.207  |
> | RPBG  |√| 22.39  | 0.735  | 0.225  |
>
> Results on ScanNet
> | Method | Per-scene | PSNR ↑ | SSIM ↑ | LPIPS ↓ |
> |------:|:--------------:|-------:|-------:|--------:|
> | 3DGS  | √              | 31.29  | 0.896  | 0.219   |
> | RPBG  | √              | 31.10  | 0.885  | 0.171   |
> | Ours | ×           | **31.62** | **0.900** | **0.139** |
>
> Results on DTU
> | Method | Per-scene  | PSNR ↑ | SSIM ↑ | LPIPS ↓ |
> |------:|:--------------:|-------:|-------:|--------:|
> | 3DGS  | √              | 24.92  | 0.871  | 0.191   |
> | RPBG  | √              | 23.80  | 0.822  | 0.173   |
> | Ours | ×           | **28.33** | **0.921** | **0.124** |

---

> ### Author Response · Authors · 2025-11-28
> **Response to Reviewer Vaw3 (2/3)**
>
> Results on Thuman2.0
> | Method | Per-scene | PSNR ↑ | SSIM ↑ | LPIPS ↓ |
> |------:|:--------------:|-------:|-------:|--------:|
> | 3DGS  | √              | **41.57**  | **0.989**  | **0.004**   |
> | RPBG  | √              | 36.60  | 0.982  | 0.020   |
> | Ours | ×           | 41.21 | 0.988 | **0.004** |
>
> It is worth noting that both 3DGS and RPBG adopt per-scene optimization paradigms—3DGS fits Gaussian primitives, whereas RPBG optimizes neural point descriptors—to each individual training scene for high-quality novel view synthesis. In contrast, **our method is trained across diverse scenes and generalizes to previously unseen environments, providing greater flexibility for practical application.**
>
> [1] Qingtian Zhu, Zizhuang Wei, Zhongtian Zheng, Yifan Zhan, Zhuyu Yao, Jiawang Zhang, Kejian Wu, and Yinqiang Zheng. RPBG: Towards Robust Neural Point-Based Graphics in the Wild. In Proceedings of the European Conference on Computer Vision (ECCV), pp. 389–406, 2024.
>
> **Comment 3.** The rendering fps is not suitable for real-time applications, though it's not considered as a drawback of this work.
>
> **Response.** We thank the reviewer for the insightful comment and agree that the rendering FPS of the current implementation is lower than real-time rates. However it is also worth noting that, **compared to the evaluated baselines, our method achieves the best rendering quality while maintaining a favorable trade-off in rendering efficiency**, as reported in Appendix Table 19 of the manuscript. For clarity, we summarize the quantitative results on DTU using input point clouds of 1 million points, with output images rendered at a resolution of 640 × 512 pixels.
>
> Evaluation of quality (PSNR)-efficiency trade-off on the DTU (640 × 512).
> | Method     | PSNR ↑ | SSIM ↑ | LPIPS ↓ | GPU hrs ↓ | FPS ↑  |
> |-----------:|------:|-------:|--------:|----------:|-------:|
> | NPBG++     | 22.32 | 0.833  | 0.327   | ∼10       | **11**     |
> | TriVol     | 20.02 | 0.674  | 0.483   | ∼48       | 0.07   |
> | PFGS       | 25.44 | 0.901  | 0.164   | ∼40       | 0.5    |
> | DiffPBR-E  | 28.15 | 0.919  | 0.138   | **∼8**        | 9.75   |
> | DiffPBR-Q  | **28.45** | **0.935**  | **0.124**   | **∼8**        | 1.9    |
>
> Note that, due to the lightweight and highly efficient design of the CoNo-Splatting module, the rendering efficiency of our method is primarily constrained by the output resolution of the refinement diffusion network, as illustrated in Figure 7 of the manuscript.
>
> We thank the reviewer once again for the insightful comment. As an important factor for real-time applications, we will consider improving rendering efficiency as a key direction for future work. We have also included the following statement under section "G LIMITATIONS AND FUTURE DIRECTIONS": *"Besides, the rendering FPS of the current implementation is lower than real-time rates, which is mainly constrained by the output resolution."*
>
> **Comment 4.** What does the PyTorch3D in the experiments particularly mean? Do the authors mean directly rasterizing points by PyTorch3D? If so, it shouldn't be a baseline method since PyTorch3D is just an implementation.
>
> **Response.** We thank the reviewer for the careful reading and for raising this question. In our experiments, **PyTorch3D here refers to converting the input 3D points into points with a fixed radius and performing rasterization to the target view without further refinement.** Specifically, we adopt the native point cloud renderer provided in the open-source PyTorch3D library, as introduced in "Accelerating 3D Deep Learning with PyTorch3D".
>
> While such rasterizers follow geometrically correct projection rules, they often struggle when handling sparse regions, holes, or noisy points. Learning-based approaches, including ours, are designed to mitigate these issues, which makes PyTorch3D a relevant reference point for contextualizing our design choices.
>
> **Comment 5.** What if Adaptive CoNo-Splatting is applied to other methods? I wonder its performance as a plug-and-play module.
>
> **Response.** We thank the reviewer for this insightful comment. To evaluate the performance of our CoNo-Splatting module, we incorporate it into PFGS, one of the baseline methods in our experiments. The quantitative results are summerized below:
> | Method              | PSNR ↑ | SSIM ↑ | LPIPS ↓ | Training (h) | Inference (fps) |
> |--------------------|-------:|-------:|--------:|-------------:|----------------:|
> | PFGS               | 19.86  | 0.758  | 0.452   | 5 + 36        | 3.6             |
> | PFGS + CoNo-Splatting | **21.05** | **0.782** | **0.433** | **2 + 17**      | **4.5**         |

---

> ### Author Response · Authors · 2025-11-28
> **Response to Reviewer Vaw3 (3/3)**
>
> Concretely, PFGS first employs a feature extractor to obtain point-wise features and uses a Gaussian attribute regressor to predict per-point parameters such as scale and opacity. The predicted point colors and features are then rasterized into the target view for subsequent refinement. In this experiment, we discard its Gaussian regressor and the associated rendering pipeline, and replace them with our adaptive point-based splatting module. The results demonstrate that this replacement significantly improves both rendering quality and model efficiency. **First, this replacement brings a reduction in training and inference time due to replacing the convolutional layers in the Gaussian regressor with our adaptive numerical design. Moreover, it also reduces the training gap between the two stages in orignal PFGS, leading to better overall model performance.**
>
> **Comment 6.** Also it seems that the authors have misused the tex commands of \citep{} and \citet{}. Please correct these citation formats in the modified manuscript accordingly.
>
> **Response.** We thank the reviewer for the careful reading and valuable feedback, which have helped improve the quality of our manuscript. Following your suggestion, we have corrected the misused citation formats in the revised manuscript.

---

### Official Review · Reviewer_6AMT · 2025-10-29

**Soundness:** 2
**Presentation:** 2
**Contribution:** 2
**Rating:** 4
**Confidence:** 3

**Summary:**

This paper proposes DiffPBR, a point-based rendering framework that integrates a diffusion model to enhance the quality and view consistency of images rendered from colored point clouds. The method consists of three main components: 1) An adaptive "CoNo-Splatting" technique that rasterizes both color and a structured 3D noise map from the point cloud; 2) A residual diffusion process that learns to predict the difference between the coarse rendering and the ground truth, conditioned on the rendered noise and a mask; 3) A learnable global scale parameter optimized via a balance of coverage and compactness losses. The authors demonstrate state-of-the-art results on several datasets (ScanNet, DTU, THuman2.0) in terms of PSNR/SSIM/LPIPS, while also reporting significant improvements in training efficiency and inference speed compared to the baseline PFGS.

**Strengths:**

The overall framework is well-motivated and addresses the core artifacts of point-based rendering (holes, aliasing, view inconsistency) in a unified, learnable manner. The integration of a diffusion model for refinement is a logical and powerful choice.

**Weaknesses:**

1. Limited Conceptual Novelty: The core components—"residual diffusion" and "3D-consistent noise"—are applications of existing ideas rather than fundamental innovations. Residual learning is standard in image restoration, and 3D-consistent generation is a well-established research direction. The paper does not sufficiently delineate its conceptual advance beyond the specific application to point cloud rendering.
2. Inadequate Comparison to True SOTA: The most significant omission is a direct comparison with 3D Gaussian Splatting (3DGS) and its variants, which are the current undisputed state-of-the-art in terms of rendering quality and speed for scene-specific novel view synthesis. Bypassing this comparison undermines the claim of superior performance.
3. Unconvincing and Unfair Efficiency Claims: The dramatic reduction in training time compared to PFGS is likely attributable to the replacement of PFGS's heavy CNN-based point refinement network with a much simpler rendering front-end. This is an architectural choice, not necessarily a pure algorithmic efficiency gain, making the comparison somewhat misleading.
4. Methodological Ambiguities: The end-to-end joint training of the splatting parameters (β) and the diffusion model creates a non-stationary training target for the diffusion network, the stability and necessity of which are not validated through ablations. The use of the term "adversarial balance" for a simple weighted sum of losses is inaccurate and misleading.

**Questions:**

1. Why was a direct quantitative and qualitative comparison with 3D Gaussian Splatting (3DGS) omitted? Given that 3DGS represents the current peak performance for this problem (even if scene-specific), how does DiffPBR's rendering quality on a per-scene basis compare?
2. The joint optimization of the splatting stage (via $L_ens$) and the diffusion model (via $L_dm$) is non-standard. Can you provide an ablation study training the diffusion model with a fixed β (e.g., from a pre-trained stage or a heuristic) to demonstrate that the joint training is truly beneficial and not detrimental to convergence?

---

> ### Author Response · Authors · 2025-11-28
> **Response to Reviewer 6AMT (1/3)**
>
> We sincerely thank Reviewer 6AMT for the time and effort in reviewing our work and providing constructive feedback. We address each concern or question below.
>
> **Comment 1** Clarification of our contributions.
>
> **Response.** We thank the reviewer for this comment. We would like to clarify that our paper tackles an important and highly active research problem, and our method achieves state-of-the-art performance on this task. **Our contributions introduce several technical innovations, and the empirical results consistently demonstrate large improvements over strong baselines.** Therefore, we respectfully disagree with the assessment that our work lacks conceptual novelty or fundamental contributions. We believe such a characterization does not accurately reflect the significance of our technical advances nor the measurable impact validated by our experiments.
>
> Moreover, we clarify that maintaining 3D consistency remains an active and important research direction, and many works has proposed leveraging 3D spatial information to improve rendering coherence. For example, baselines such as PFGS and NPBG++ rely on **heavy feature extractors** to aggregate spatial features from input point clouds or images. In contrast, **we provide a simple and elegent way by introducing 3D-consistent noise to guide the residual diffusion refinement**.  Compared to PFGS, our design not only achieves substantially faster training (**8 hours vs. 41 hours**) but also yields higher reconstruction quality (**4~5 dB PSNR improvement**) and better multi-view consistency (**11.52% average improvement in TSED**).
>
> Regarding the concern about the novelty of 3D-consistent noise, we respectfully disagree with the statement that our 3D consistent noise is simple reuse of prior technique, and the reviewer provides no citation to substantiate this claim. We acknowledge that a concurrent work, GSD [1] apdots a similar idea in the text-to-3D generation task. However, their problem setting is fundamentally different from ours, and **their strategy upsamples noise in point-cloud space, which disrupts the pixel-level alignment between the rendered RGB image and the noise field, resulting in blurry refinement artifacts**—an effect demonstrated in Figure 9 and Table 10 of our manuscript. To further clarify, we provide ablated comparisons as follows:
> | Strategy | Noise Pos. | Noise Col. | PSNR↑ | SSIM↑ | LPIPS↓ |
> |----------|-------------|------------|------|-------|--------|
> | 1        | ✗           | ✓          | **41.27** | **0.988** | **0.004** |
> | 2        | ✓           | ✓          | 40.02 | 0.985 | 0.010 |
>
> These results highlight that our 3D-consistent noise formulation is specifically designed to the point cloud rendering setting, and its design is essential for achieving high-fidelity image synthesis without the artifacts introduced by alternative approaches.
>
> Based on this, we integrate it into our Spatial-aware Residual Diffusion pipeline. **With stable noise guidance across views, our method achieves substantial improvements in multi-view consistency**, as evidenced by the TSED metric reported in the table below.
>
> | T  | Ours | PFGS   | NPBG++ | TriVol | 3DGS   | RPBG   |
> |---:|---------:|-------:|-------:|-------:|-------:|-------:|
> | 2  | **0.2347** | 0.2143 | 0.1837 | 0.1531 | **0.2347** | 0.2245 |
> | 4  | **0.4796** | 0.4286 | 0.3776 | 0.3265 | 0.4694 | 0.4388 |
> | 8  | **0.8776** | 0.7755 | 0.7143 | 0.6735 | 0.8571 | 0.8061 |
>
> Specifically, the TSED value under smaller $T$ primarily reflects the temporal flickering or instability between consecutive output frames, while under larger values of $T$, it corresponds to the overall consistency of the output frames across large camera view deviations.
>
> Overall, we believe that our method is carefully designed to form an effective framework for point-based neural rendering, providing a new angle for solving the core challenges of the task. We hope this clarification addresses the reviewer’s concerns, and we would be glad to provide additional details if further explanation is required.
>
> [1] Min-Seop Kwak, Donghoon Ahn, Inès Hyeonsu Kim, Jin-Hwa Kim, and Seungryong Kim. Geometry-aware score distillation via 3d consistent noising and gradient consistency modeling. arXiv preprint arXiv:2406.16695, 2024
>
> **Comment 2.** Rendering Performance compared with 3DGS.
>
> **Response.** We thank the reviewer for this comment. We would like to clarify that 3DGS is specifically designed for per-scene optimization, whereas our method adopts a generic training paradigm over a diverse set of scenes. **Once trained, our model can be directly applied to unseen scenes in a feed-forward manner without scene-specific optimization, which fundamentally distinguishes our approach from 3DGS**.

---

> ### Author Response · Authors · 2025-11-28
> **Response to Reviewer 6AMT (2/3)**
>
> To further address the reviewer's concern, we have provided a comprehensive comparison and detailed discussion in Appendix E.1 of the revised manuscript, and summarize the quantitative results here for clarity.
>
> Results on ScanNet
> | Method | Per-scene | PSNR ↑ | SSIM ↑ | LPIPS ↓ |
> |------:|:--------------:|-------:|-------:|--------:|
> | 3DGS  | √              | 31.29  | 0.896  | 0.219   |
> | RPBG  | √              | 31.10  | 0.885  | 0.171   |
> | Ours | ×           | **31.62** | **0.900** | **0.139** |
>
> Results on DTU
> | Method | Per-scene  | PSNR ↑ | SSIM ↑ | LPIPS ↓ |
> |------:|:--------------:|-------:|-------:|--------:|
> | 3DGS  | √              | 24.92  | 0.871  | 0.191   |
> | RPBG  | √              | 23.80  | 0.822  | 0.173   |
> | Ours | ×           | **28.33** | **0.921** | **0.124** |
>
> Results on Thuman2.0
> | Method | Per-scene | PSNR ↑ | SSIM ↑ | LPIPS ↓ |
> |------:|:--------------:|-------:|-------:|--------:|
> | 3DGS  | √              | **41.57**  | **0.989**  | **0.004**   |
> | RPBG  | √              | 36.60  | 0.982  | 0.020   |
> | Ours | ×           | 41.21 | 0.988 | **0.004** |
>
> Due to the per-scene training manner of 3DGS and RPBG, we obtain the quantitative results on several scenes. Note that in DTU, the absence of background points often leads 3DGS and RGBP to produce blurry regions and other artifacts in the background, which degrades the overall rendering quality.
>
> **Comment 3.** Relation to efficiency clarification.
>
> **Response.** We thank the reviewer for this comment. The training efficiency of our method comes from two main aspects:
> First, we argue that the previous methods such as PFGS uses a heavy CNN-based regressor to obtain 3D consistent point feature, requiring a time-consuming training stage. In contrast to the heavy way using network to maintain the 3d consistent, **we use a lightweight and efficient numerical strategy by adding Gaussian noise**. After adaptive splatting, the rendered image naturally preserves rich 3D structural cues, allowing our refinement network to remain simple.
>
> To further prove this point, we adopt one of our contributions, *Adaptive CoNo-Splatting* module to PFGS in a plug-and-play fashion. Specifically, we remove its Gaussian regressor and the corresponding rendering pipeline, and replace them with our *Adaptive CoNo-Splatting* module. The quantitative comparison is summarized below:
> | Method              | PSNR ↑ | SSIM ↑ | LPIPS ↓ | Training (h) | Inference (fps) |
> |--------------------|-------:|-------:|--------:|-------------:|----------------:|
> | PFGS               | 19.86  | 0.758  | 0.452   | 5 + 36        | 3.6             |
> | PFGS + CoNo-Splatting | **21.05** | **0.782** | **0.433** | **2 + 17**      | **4.5**         |
>
> Notably, **both rendering quality and model efficiency improve, with the training time reducing by more than half**. This improvement stems from replacing the heavy convolutional design in Gaussian regressor with our adaptive network-free design, which simplifies the training pipeline and reduces computational overhead.
>
> Second, we fully exploit the information already encoded in the splatted image and adopt a *Spatial-Aware Residual Diffusion* paradim that predicts the residual between the splatted image and the target image. Empirically, this formulation results in noticeably faster convergence.
>
> **Comment 3.** Relation to the effect of end-to-end joint training the splatting parameters (β) and the diffusion model.
>
> **Response.** We thank the reviewer for this comment. In our implementation, the two complementary regularizers: $\mathcal{L} _ {cmp}$ encourages smaller $\beta$ to reduce artifacts, while $\mathcal{L} _ {cov}$ encourages larger $\beta$ to cover more pixels. These two losses act in opposite directions, creating a natural balance that prevents $\beta$ from drifting excessively, which is also the reason we originally referred to their combined effect as an "adversarial balance". We apologize for any potential misunderstanding caused by inaccurate terminology and have revised the wording in the updated manuscript.
>
> To address the reviewer's concern on their effect on training stability, particularly under fixed $\beta$, we provide quantitative results as follows:
>
> | Method                      | Splatting PSNR ↑ | Refinement PSNR ↑ |
> |----------------------------:|----------------:|------------------:|
> | (a) KNN-based scale calc.   | 13.43           | 22.05             |
> | (a) + (b) adaptive scale reg.| **15.14**       | **23.28**         |
>
>
> | Setting     | PSNR ↑ | SSIM ↑ | LPIPS ↓ |
> |------------:|-------:|-------:|--------:|
> | w/o $L _ {cns}$    | 22.31  | 0.809  | 0.418   |
> | w/o $L _ {cov}$    | 22.69  | 0.815  | 0.414   |
> | w/o $L _ {cmp}$    | 22.97  | 0.820  | 0.404   |
> | Full    | **23.28** | **0.827** | **0.399** |

---

> ### Author Response · Authors · 2025-12-02
> **Response to Reviewer 6AMT (3/3)**
>
> The diffusion model using KNN-based heuristic scale calculation, with fixed $\beta$ value, yields refinement with PSNR equals to 22.05. **With adaptive scale regulation, the results improve by a PSNR of 1.23**. Moreover, a comparison between these two Tables shows that optimizing $\beta$ with the diffusion loss alone is inefficient, yielding only a 0.26 dB improvement over the heuristic KNN initialization. **Incorporating both regularizers, however, substantially improves rendering fidelity, achieving a PSNR gain of 0.97 compared to using only the diffusion loss.**

---

### Official Review · Reviewer_Z48r · 2025-10-30

**Soundness:** 3
**Presentation:** 3
**Contribution:** 2
**Rating:** 6
**Confidence:** 4

**Summary:**

This paper proposes a model, called DiffPBR, that generates photo-realistic, view-consistent point-cloud renderings. DiffPBR is a two-stage method that combines adaptive, noise-consistent splatting with a diffusion model to render point clouds from calibrated camera view parameters. First, the point cloud is projected into the image domain using point-based splatting with a globally truncated scatter-distance threshold. This step introduces noise and artificial rasterization artifacts, serving as an initial rendering proposal. Then, a DDPM flow-based backbone is trained end to end to produce denoised, photo-realistic renderings for each camera view.

To show that DiffPBR is an end-to-end solution for rendering single-modal point-cloud input, the authors conduct experiments on the ScanNet, DTU, and THuman 2.0 datasets to demonstrate (i) improved rendering quality and (ii) manageable rendering speed (**2**–10 FPS, depending on Quality mode or Efficient mode, which the authors refer to as DiffPBR-Q and DiffPBR-E, respectively).

**Strengths:**

The paper offers an insightful focus on decoupling the point cloud information from the rendering pipeline’s noise scheduling. Point-based splatting is much lighter than PFGS’s per-point CNN scale predictor and is explicitly tuned to produce good inputs for the diffusion refiner rather than the final image.

The idea of adopting a diffusion model that diffuses from something other than pure Gaussian noise recalls [Cold Diffusion](https://proceedings.neurips.cc/paper_files/paper/2023/file/80fe51a7d8d0c73ff7439c2a2554ed53-Paper-Conference.pdf), which might indicate its efficacy for this particular problem.

The experiments are self-contained, and the ablations are extensive.

**Weaknesses:**

My biggest concern is the training setup given what is already known about the input. The assumption that the point cloud and calibrated camera views are available is both an advantage and a restriction. Is DiffPBR aware of noise arising from point-cloud registration, and if it is trained per scene, why is it not biased, and how can it transfer or generalize across scenes? I understand Figure 5 tries to address transferability by directly migrating models without fine-tuning, but I am unsure why such a scheme would work. More experiments may be needed to convince the reader that geometry-aware noise patterns are interchangeable.

In my view, recent works like [Vanilla Gaussian Splatting](https://repo-sam.inria.fr/fungraph/3d-gaussian-splatting/), [FSGS](https://zehaozhu.github.io/FSGS/), and [RPBG](https://www.ecva.net/papers/eccv_2024/papers_ECCV/papers/02379.pdf) should be included for comparison even though they are NVS methods that rely on (sparse) multi-view input. Comparing to their results would provide supportive evidence for why DiffPBR can outperform and better utilize point-cloud information. My suggestion is to test bilaterally: give one NVS method a full prior of point-cloud information, and give DiffPBR the sparse-view COLMAP registration results in the large-scale scenarios used in those NVS papers. These comparisons would make the paper more compelling.

Some minor concerns, if resolved, would also improve the completeness of the paper, and one could consider raising the score if most concerns are addressed or justified.

* No code is provided for reproducibility.
* L204–206, Eqn. (1): using the median of all average scales seems a very bold guess. What happens if a different percentile (the median is the 50th percentile) is used, and why is the mean not an appropriate statistic?
* In L283 A1 L11, why is the residual ground truth given as a linear interpolant of $I_{\epsilon}^v$ and $I_{r}^v$? Can you explain $\gamma_t$, as it seems undefined.
* Why do DiffPBR-E and DiffPBR-Q require the same training time on average per scene? I think their training pipelines can be interchangeable, and DiffPBR-E may be more generalizable since it uses a relative image-consistency loss rather than comparing with ground truth. Have you considered cross-checking whether DiffPBR-Q training plus one-step inference would work?
* I notice that in Table 2, PFGS is reported to have 3.6 FPS while DiffPBR-Q only gets 2 FPS inference speed. Please clarify the claim in L24 of the abstract.
* Although the ablation is extensive, the choices of $\lambda_{cov}=0.01$ and $\lambda_{cmp}=0.1$ in L864 are not discussed. Do they affect performance?

**Questions:**

* Can you define the CNSplat function (L246–L248) as a mathematical formula?
* In L49, does FPGS refer to PFGS?
* Can you explain why a graphics-based rendering technique can fail to provide photorealistic renderings on the given datasets? Have you considered rendering the ScanNet or DTU data using Mitsuba rather than PyTorch3D?
* In L714, should $\mathcal{F}$ be $\mathcal{F}_{\theta}$?
* Apart from Figure 6 (noise-covariance patterns across views) and average rendering performance, have you considered other schemes to quantify view consistency of the rendering output? I don’t have an immediate answer in mind, but I would like to hear more about how to safeguard “consistent multi-view synthesis” (L93).

---

> ### Author Response · Authors · 2025-11-28
> **Response to Reviewer Z48r (1/4)**
>
> We sincerely thank Reviewer Z48r for the time and effort in reviewing our work and providing supportive comment.
>
> **Comment 1.** How can DiffPBR transfer or generalize across different scenes?
>
> **Response.** We thank the reviewer’s detailed comment. We would like to kindly clarify that **our method is not designed for per-scene training. Instead, it is a feed-forward model trained across diverse datasets and generalizes well to previously unseen scenes.** Specifically, our *Spatial-Aware Residual Diffusion* serves as a post-refiner for point-rasterized images, which predicting the *residual* between the rasterized color image and the ground-truth image as well as a structured noise. During training, the model observes diverse scenes and gradually learns to correct subtle geometric distortions and recover missing high-frequency details in the rendered images. At test time, the color and geometry-aware noise features are re-initialized for each unseen point cloud, while the degradation characteristics introduced by point-based rasterization (e.g., holes and fluttering artifacts) remain structurally similar across scenes. This consistency enables the trained model to reliably refine new inputs and produce faithful, high-quality renderings.
>
> **Comment 2.** Evaluation of the multi-view synthesis.
>
> **Response.** We thank the reviewer for the constructive comment. To evaluate the view-consistency of our model, we add a quantitative evaluation using the Thresholded Symmetric Epipolar Distance (TSED) metric [1], which quantifies the number of consistent frame pairs in a sequence. We compare with both generalizable methods and per-scene optimized method as follows:
>
> | T  | Ours | PFGS   | NPBG++ | TriVol | 3DGS   | RPBG   |
> |---:|---------:|-------:|-------:|-------:|-------:|-------:|
> | 2  | **0.2347** | 0.2143 | 0.1837 | 0.1531 | **0.2347** | 0.2245 |
> | 4  | **0.4796** | 0.4286 | 0.3776 | 0.3265 | 0.4694 | 0.4388 |
> | 8  | **0.8776** | 0.7755 | 0.7143 | 0.6735 | 0.8571 | 0.8061 |
>
> Specifically, the TSED value under smaller $T$ primarily reflects the temporal flickering or instability between consecutive output frames, while under larger values of $T$, it corresponds to the overall consistency of the output frames across large camera view deviations. For better intuitive evaluation, we recommend the reviewer refer to the supplementary video for clearer visualizations and comparisons.
>
> [1] Jason J. Yu, Fereshteh Forghani, Konstantinos G. Derpanis, and Marcus A. Brubaker. Long-term photometric consistent novel view synthesis with diffusion models. In *Proceedings of the IEEE/CVF International Conference on Computer Vision and Pattern Recognition*, pp. 7094–7104, 2023.
>
> **Comment 3.** Point clouds derived from real-world scans inevitably contain noise from registration. How does DiffPBR handle such geometric imperfections, and why is the median used for heuristic scale initialization instead of other percentiles or the mean?
>
> **Response.** We thank the reviewer’s insightful comment. As pointed out, noise introduced during registration, sensing, or estimation can make point-based rasterization susceptible to artifacts such as holes and aliasing. The point clouds used in our main experiments (DTU and ScanNet) are derived from real-world scans and therefore inherently contain such structural noise.
>
> To address this, **our method combines a robust heuristic initialization with a data-driven neural refinement strategy**. During the initialization of point scales, we begin by calculating a heuristic scale based on K-Nearest Neighbor ($\text{KNN}$) distances. Recognizing that these distances are easily skewed by outliers in real point clouds, we introduce a global truncation strategy, where the median provides a robust statistic compared with the mean, and alternative percentiles behave similarly as long as they suppress outlier effects. After that, we feed the projection of point clouds into a neural refinement network. Trained on a large-scale dataset with diverse scenes, the network encounters various noise patterns and learns noise-resistant rendering priors. This data-driven refinement equips our method with noise awareness and robustness, effectively mitigating artifacts caused by imperfect geometric inputs.
>
> To further address the reviewer’s concern, we add different levels of noise, controlled by a variance factor  $\sigma$ , to the clean input point cloud, and evaluate the robustness of our method under these degraded conditions. For clarify, we provide the quantitative results (PSNR) as follows:
>
> | Method | $\sigma$=1e-3 | $\sigma$=2e-3 | $\sigma$=3e-3 | Variance $\downarrow$|
> |-----------:|---------------:|---------------:|---------------:|-----------:|
> | PFGS       | 33.80          | 33.09          | 32.14          | 1.89       |
> | DiffPBR-E  | 39.62          | 39.09          | 38.60          | 0.73       |
> | DiffPBR-Q  | **40.33**      | **39.62**      | **39.05**      | **0.69**   |

---

> ### Author Response · Authors · 2025-11-28
> **Response to Reviewer Z48r (2/4)**
>
> Moreover, we also conducted an additional ablation study on the heuristic scale initialization. Specifically, we compare four strategies: 25th percentile, 50th percentile (median), 75th percentile, and Mean, respectively. We provide a comprehensive comparison and detailed discussion in Appendix E.1, particularly the sub-section titled "Heuristic Scale Initialization". We summarize the quantitative results here for clarity.
>
> | Metric |   Q1 (25% )   |   Q2 (50% Median)   |   Q3 (75%)   |   Mean  |
> |-------:|-------:|---------------:|-------:|--------:|
> | PSNR   | 22.41  | **23.28**          | 22.91  | 23.19   |
> | SSIM   | 0.803  | **0.827**          | 0.819  | 0.822   |
> | LPIPS  | 0.421  | **0.399**          | 0.412  | 0.404   |
>
> In our experiments, the median (Q2) performs slightly better than other statistics. **While both mean and median are commonly used, the median is more robust to outliers and provides stronger filtering, making it better suited for our global truncation strategy.** Therefore, we adopt the median in our method.
>
> **Comment 4.** Recent works such as Vanilla Gaussian Splatting, FSGS, and RPBG should be included for comparison. My suggestion is to test bilaterally: give one NVS method a full prior of point-cloud information, and give DiffPBR the sparse-view COLMAP registration results in the large-scale scenarios.
>
> **Response.** We thank the reviewer for this valuable suggestion. We would like to clarify that, because **our method follows a generic training paradigm rather than per-scene optimization as in 3DGS**, we did not include 3DGS as a baseline in the previous submission. To further address the reviewer's concern, **we have now conducted additional comparative experiments with per-scene optimized methods**. Specifically, we choose two NVS methods 3DGS and RPBG for comparison and conduct experiments on the large scale unbounded dataset Tanks and Temples.
>
> We provide a comprehensive comparison and detailed discussion in Appendix E.1, and summarize the quantitative results here for clarity.
> Sparse point cloud from COLMAP (Panther scene)
> | Method | PSNR ↑ | SSIM ↑ | LPIPS ↓ |
> |--------|-------:|-------:|--------:|
> | Ours   | **22.97**  | **0.783**  | **0.126**   |
> | 3DGS   | 22.16  | 0.726  | 0.300   |
> | RPBG   | 22.30  | 0.746  | 0.272   |
>
> Dense point cloud from Pi3[1] (Panther scene)
> | Method | PSNR ↑ | SSIM ↑ | LPIPS ↓ |
> |--------|-------:|-------:|--------:|
> | Ours   | **23.49**  | **0.854**  | **0.109**   |
> | 3DGS   | 22.68  | 0.754  | 0.214   |
> | RPBG   | 22.93  | 0.780  | 0.181   |
>
> Note that we obtain the dense point cloud using the a feed-forward regressing model Pi3. The results demonstrate that our method achieves the best performance among all compared baselines under both sparse and dense settings.
>
> [1] Yifan Wang, Jianjun Zhou, Haoyi Zhu, Wenzheng Chang, Yang Zhou, Zizun Li, Junyi Chen, Jiangmiao Pang, Chunhua Shen, and Tong He. $\Pi^3$: Permutation-Equivariant Visual Geometry Learning. arXiv preprint arXiv:2507.13347, 2025.
>
> **Comment 5.** No code is provided for reproducibility.
>
> **Response.** We appreciate the reviewer’s attention. Our main paper and supplementary material provide detailed algorithmic steps (Algorithms 1–4) for reproducibility, and we will make the code publicly available upon acceptance.
>
> **Comment 6.** In L283 A1 L11, why is the residual ground truth given as a linear interpolant of  $I_\epsilon^t$   and  $I_r^t$ ? Can you explain $\gamma_t$, as it seems undefined.
>
> **Response.** We thank the reviewer for raising this thoughtful question. **The additive operation is performed in the noise space, where image representations behave as vectors that can be linearly combined**. In this space, $I_r^v=I_c^v-I_0^v$ forms a directional vector pointing from the clean image toward the rendered image, enabling the initialization $I_\epsilon^v+I_r^v$ to guide the reverse diffusion along this residual direction rather than starting from unguided noise. Regarding $\gamma_t$, it is a compact notation defined as $ \gamma_t = (1 - \sqrt{\bar{\alpha}_t})\cdot\sqrt{1-\bar{\alpha}_t}$  as specified below Algorithm 2 (Multi-step DiffPBR-Q Inference). We have revised the manuscript to restate this definition in the Appendix A to ensure it can be easily located.

---

> ### Author Response · Authors · 2025-11-28
> **Response to Reviewer Z48r (3/4)**
>
> **Comment 7.** What is the difference in training time and generalization ability between DiffPBR-Q and DiffPBR-E, and does one-step DiffPBR-Q work?
>
> **Response.** We thank the reviewer for this insightful comment. For both variants of DiffPBR, **the forward pass time per diffusion iteration is essentially identical, as they share the same network architecture. Consequently, the total training time mainly depends on the number of training iterations**, which we set to be the same for both variants to ensure a fair comparison, and this choice has been empirically effective. Due to their different training objective, DiffPBR-Q is naturally more generalizable compared to DiffPBR-E across diverse scenes. There might be a typo from in your text that *"DiffPBR-E may be more generalizable ..."* should refer to DiffPBR-Q.
>
> Moreover, as your insightful suggestion, we have conducted cooresponding experiments as presented in Table 9, where  $\epsilon+S.$  denotes DiffPBR-Q training plus one-step inference. The results suggest that: $\epsilon$ -prediction benefits more from multi-step inference, while x0-prediction performs slightly better with one-step inference. For clarify, we provide part of the results on ScanNet as follows:
>
> | Method |  PSNR↑ |  SSIM↑  | LPIPS↓ |
> |-------:|-------:|-------:|------:|
> | ϵ + M.  | **23.28** | **0.827** | **0.399** |
> | ϵ + S.  | 20.52 | 0.790 | 0.436 |
> | x₀ + M. | 20.88 | 0.793 | 0.428 |
> | x₀ + S. | 22.92 | 0.824 | 0.400 |
>
> where we simplify the terms "ϵ-prediction" as "ϵ","x0-prediction" as "x0", "Multi-step" as "M.", and"Single-step" as "S.", respectively.
>
> **Comment 8.** I notice that in Table 2, PFGS is reported to have 3.6 FPS while DiffPBR-Q only gets 2 FPS inference speed. Please clarify the claim in L24 of the abstract.
>
> **Response.** We thank the reviewer for this constructive comment. The 10 FPS rendering speed mentioned in the abstract refers to DiffPBR-E, which achieves slightly lower quality than DiffPBR-Q. We have revised the manuscript accordingly to clarify this description. The revision is as follows:
> *and increase from 3.6 FPS to 10 FPS (our one-step variant) in rendering speed.*
>
> **Comment 9.** Although the ablation is extensive, the choices of $\lambda_{cov}=0.01$  and $ \lambda_{cmp}=0.1 $ in L864 are not discussed. Do they affect performance?
>
> **Response.** We thank the reviewer for this insightful comment. As described in C.3 of the manuscript, the training loss is a weighted sum of the splatting and diffusion objectives:$\mathcal{L} = \lambda _ {cov}\mathcal{L} _ {cov} + \lambda _ {cmp}\mathcal{L} _ {cmp} + \mathcal{L} _ {dm}$ , where $\mathcal{L} _ {dm} $is the main training objective. The regularizer  $\mathcal{L} _ {cmp} $ encourages the shrinking of valid regions by reducing $ \beta $, which—within a moderate range—leads to the appearance of holes, as illustrated in Figure 2(b) of the manuscript. This behavior pushes the diffusion model to recover finer details from the degraded splatted images. In contrast,  $\mathcal{L} _ {cov}$  counteracts this degeneration by encouraging $ \beta $ to expand, thereby preventing the splatted image from collapsing, as shown in Figure 2(d) of the manuscript. The effects of these two terms are consistently demonstrated in Figure 3 and Table 6 of the manuscript. Empirically, we observe that **moderately smaller  $\beta$  values, guided by $\mathcal{L} _ {cmp} $, help the diffusion model restore finer structures, whereas overly small  $\beta$  induces bleeding artifacts due to background points incorrectly projecting to the foreground. To prevent this degeneration,  $\mathcal{L} _ {cov} $provides a counterbalancing force that stabilizes $\beta$.**
>
> To further address the reviewer’s concern, we additionally evaluate different configurations of  $\lambda _ {cmp}$ and  $\lambda _ {cov}$. A comprehensive comparison is provided in Appendix E.1 of the revised manuscript, particularly the sub-section titled "CoNo-splatting Configurations", and we summarize the quantitative results here for clarity.
>
> Effect of $\lambda _ {cmp}$ (with $\lambda _ {cov}=0$).
> | $λ _ {cmp}$       |   0    |  0.01  |  0.1   |   1    |
> |------------:|-------:|-------:|-------:|-------:|
> | PSNR↑        | 22.31  | 22.57  | **22.69**  | 22.39  |
> | SSIM↑        | 0.804  | 0.815  | **0.821**  | 0.806  |
> | LPIPS↓       | 0.424  | 0.412  | **0.406**  | 0.417  |
>
> Effect of $\lambda _ {cov}$ (with $\lambda _ {cmp}=0$).
> | $λ _ {cov}$       |   0    | 0.01   |  0.1   |   1    |
> |------------:|-------:|-------:|-------:|-------:|
> | PSNR↑        | 22.31  | **23.01**  | 22.97  | 22.62  |
> | SSIM↑        | 0.804  | **0.826**  | 0.822  | 0.810  |
> | LPIPS↓| 0.424  | **0.402**  | 0.407  | 0.419  |

---

> ### Author Response · Authors · 2025-11-28
> **Response to Reviewer Z48r (4/4)**
>
> **Comment 10.** Can you define the CNSplat function (L246–L248) as a mathematical formula?
>
> **Response.** We thank the reviewer for this thoughtful question. Below we provide a formal and renderer-agnostic mathematical definition of CNSplat, as requested.
> As described in the manuscript, we define the input point set as: $P_{in}=\{(\mathrm{x _ i}, \mathrm{c _ i}, \mathrm{\epsilon _ i}, \mathrm{s_i})\} _ {i=1}^n, $ where $\mathrm{x _ i}\in\mathbb{R}^3$ is the 3D position, $\mathrm{c _ i}\in\mathbb{R}^3$ is the color, $\mathrm{\epsilon _ i}\in\mathbb{R}^3$ is the noise vector, and $\mathrm{s _ i}=s_i\cdot(1,1,1)^T\in\mathbb{R}^3$ is the point scale. The color and noise are concatenated into a unified point feature  $\mathrm{f _ i} = \mathrm{concat}(c _ i, \epsilon _ i) \in \mathbb{R}^{6}. $
>
> Given the camera intrinsics $\mathrm{K^v}$, extrinsics $\mathrm{M^v}$, the forward splatting operation in CNSplat follows a general point-based rendering formulation:
>
> $F(p)=
> \frac{
> \sum_{i=1}^n
> \kappa\!\Big((p-\pi(\mathrm{K^v},\mathrm{M^v},\mathrm{x_i}))/s_i\Big)\,
> v(z_i)\, \mathrm{f_i}
> }{
> \sum_{j=1}^n
> \kappa\!\Big((p-\pi(\mathrm{K^v},\mathrm{M^v},x_j))/s_j\Big)\,
> v(z_j) + \delta},
> \qquad \forall\, p \in \Omega$
>
> where \(p\) denotes a pixel location on the feature plane $\Omega\in\mathbb{R}^6$, $\pi(\cdot)$ denotes the standard pinhole projection mapping a 3D point to its 2D image coordinate, $\kappa(\cdot)$ is a differentiable splatting kernel that distributes a point’s contribution to its neighboring pixels, $z _ i$ is the depth of point $i$ in the camera coordinate system, $v(\cdot)$ is a generic differentiable visibility weighting function, and $\delta>0$ ensures numerical stability.
>
> Finally, the color and noise maps are extracted from the rendered feature:
>
> $I _ c^v(p) = [F(p)] _ {1:3}, \quad
> I _ {\epsilon}^v(p) = [F(p)] _ {4:6}, \qquad \forall\, p \in \Omega.$
>
> In summary, the overall process can be compactly expressed as：
>
> $I _ c^v,I _ {\epsilon}^v = \mathrm{CNSplat}(\mathrm{\mathcal{P _ {in}} | K^v,M^v}) $
>
> Notably, different point-based renderers may instantiate the splatting kernel $\kappa(\cdot)$ (e.g., Gaussian, elliptical, or learned kernels) or visibility function $v(\cdot)$ (e.g., soft-z compositing, or accumulated transmittance) in different ways. Nevertheless, the abstract functional form of CNSplat remains unchanged, enabling our method to be compatible with general differentiable point-based renderers.
>
> **Comment 11.** Can you explain why a graphics-based rendering technique can fail to provide photorealistic renderings on the given datasets? Have you considered rendering the ScanNet or DTU data using Mitsuba rather than PyTorch3D?
>
> **Response.** We thank the reviewer for raising this important point. In our experiments, the input point clouds are pre-downsampled to relatively sparse densities (e.g., ~80k points for the Thuman2.0 dataset). **Due to the inherently discrete nature of point clouds, direct point-based rasterization—regardless of the specific renderer used, as long as it follows pure graphics pipeline—tends to suffer from artifacts such as holes, discontinuities, and aliasing, particularly under sparse sampling.** This limitation further underscores the necessity of a post-refinement module to achieve photorealistic point cloud rendering.
>
> To further address your concern, we have also included the rendering results from both Pytorch3D and Mitsuba in the revised manuscript under Appendix E.2, please kindly check out. For clarification, Mitsuba, being a ray-tracing-based renderer, requires explicit lighting to simulate realistic illumination, which leads to images that are visually different from PyTorch3D. While Mitsuba generally produces higher-quality images, it incurs a slower rendering time.
>
> **Comment 12.** In L49, does FPGS refer to PFGS? In L714, and in L714, should $\mathcal{F}$ be $\mathcal{F} _ \theta$?
>
> **Response.** We thank the reviewer for the careful reading. Yes — the term FPGS and \(\mathcal{F}\) were typographical errors. We have corrected them in the revised manuscript and carefully re-checked the paper for similar issues.

---

### Official Review · Reviewer_PDkh · 2025-10-31

**Soundness:** 3
**Presentation:** 3
**Contribution:** 1
**Rating:** 4
**Confidence:** 4

**Summary:**

This work presents a method for rendering 3D point clouds to images. Specifically it aims for photorealistic rendering even from sparse point-clouds. This is achieved by using a customised diffusion model to "tidy up" the sparse rendering. To ensure the diffusion results are approximately multi-view consistent, noise is first sampled supported on the point-cloud, then rasterised to different viewpoints. The model then predicts the residual to map the sparse point cloud rendering to a realistic image. On three datasets of point-clouds paired with ground-truth images, the method is shown to achieve higher-fidelity rendering than selected baselines.

**Strengths:**

The proposed pipeline is novel – the combination of size-adaptive rasterisation strategy with diffusion-based refinement using spatially (multi-view) consistent noise has not been tried before.

The approach to arranging 3D-consistent diffusion noise is simple and elegant; similarly predicting residuals instead of directly reconstructing images is a sensible choice.

The main experiments on DTU, ScanNet and THuman2 show that the proposed method achieves higher rendering fidelity than Neural Point-Based Graphics, its more-recent extension NPBG++, TriVol (based on density fields), and PFGS (based on gaussian splatting). This holds across all three datasets, according to PSNR, SSIM and LPIPS metrics.

The method is shown to be significantly faster (for training and inference) than PFGS, and also more robust to point density (i.e. it retains higher fidelity even when the point cloud becomes sparser).

There are additional experiments on cross-domain generalisation, using the fairly disparate domains of Thuman2.0 people and the DTU reconstruction dataset. Even between these very different domains, the method achieves respectable visual fidelity when trained on images from one and validated on images from the other.

There is a fairly thorough ablation study measuring the benefit of various design decisions and components of the proposed method. Most notably, the 3D-consistent noise shows a significant improvement in PSNR compared with 2D noise.

The paper is clear, well-structured, and pleasant to read.

**Weaknesses:**

The technical contribution is fairly small – diffusion with 3D-consistent noise is now fairly well-established in other tasks such as multi-view-consistent novel view synthesis. Beyond this, the task of rendering sparse point-cloud realistically has been studied by several methods (including the baselines used in this work), and the additional insight given by the present work is fairly small. Similarly the "CoNo Splatting" described in Sec 3.1 is a trivial strategy for rendering point-clouds, accompanied by a simple pair of regularisers to ensure the points are a sensible size.

The text implies that the model guarantees 3D consistency across viewpoints; however while there is a strong encouragement to do so by the 3D representation of noise, it is not guaranteed since the diffusion itself occurs in 2D pixel space.

There is no comparison against naive baselines like nearest-neighbour interpolation of colors into empty pixels in 2D image space, or off-the-shelf diffusion-based inpainting after rendering the point-cloud at one pixel per point.

The results on efficiency and robustness wrt point-cloud density both compare only with PFGS, not with the other baselines. These results should be added to give a more complete picture of relative performance.

The work ignores the fundamental issue that point-cloud rendering is ill-posed – unless we know a true physical scale for each point, there is not enough information to know whether a given set of nearby, coplanar points in fact represent a continuous, solid surface rather than some kind of holed, porous structure. Both gaussian splats and NeRFs are more explicitly 'solid' in this sense since they define densities over non-infinitesimal volumes. This point should be discussed in the introduction, which currently is presented as if there is one true

**Questions:**

See the points discussed in weaknesses above – in particular the missing baselines / ablations.

L60 "diffusion models in image restoration rely on the assumption of pure noise inputs" – this is poorly worded, they typically input both a noisy image (as conditioning for the diffusion process) and latent noise that will be conditionally decoded to the final image; thus they do not receive *only* pure noise. Moreover, it is common in image restoration to start the denoising process at a DDIM-inverted or noised version of the input image, for exactly the reasons discussed in this paragraph.

Please provide LPIPS & SSIM for the ablation and other experiments that only use PSNR – LPIPS in particular is much more reflective of perceived quality.

Many \citet or \cite should become \citep

---

> ### Author Response · Authors · 2025-11-28
> **Response to Reviewer PDkh (1/4)**
>
> We sincerely thank Reviewer PDkh for the time and effort in reviewing our work and providing constructive feedback. We address each concern or question below.
>
> **Comment 1.** Clarification on the Novelty and Technical Contributions.
>
> **Response.** We thank the reviewer for the insightful comment. We would like to make clarifications from the following perspectives:
>
> **1. Core insight of our method.** Our core insight is that **converting a deterministic sparse point cloud into a probabilistic representation better guides the refinement network**. Concretely, given a sparse point cloud, the *Adaptive CoNo-Splatting* module performs noise sampling and scale modulation, which naturally yields a probabilistic representation. When the noise points are rendered onto the image plane, the network can readily learn both deterministic and uncertain patterns, which in turn guides it to fill in missing regions in the rendered image. In contrast, conventional rendering methods, such as pyTorch3D, treat and splat each point deterministically. Once rendered, the network has limited ability to distinguish which regions require completion versus which should be preserved. Moreover, the CoNo-Splatting design brings other benifit that **the rendered noise maps provide a stable and view-consistent initialization for the reverse diffusion process**, which in turn improves multi-view consistency in the generated outputs.
>
> **2. Novelty of our 3D consistent noise.** Here, we would like to kindly clarify that the papers referenced in the Appendix are mainly focused on 3D generation tasks. They are cited as related context because they share the same high-level goal of *achieving 3D consistent* with ours, rather than as evidence that *using 3D-consistent noise* is already a well-established solution. The most related and concurrent work GSD [1] targets text-to-3D generation and therefore differs fundamentally from our point-based rendering task. Specifically, GSD maintains 3D consistency by upsampling point clouds to generate dense representations, a strategy suitable for text-driven 3D generation. However, in our rendering task, *such upsampling will disrupt the pixel-level alignment between the rendered RGB image and the noise map*, leading to blurry artifacts (Figure 11 and Table 10). This limitation necessitates a different design; our formulation avoids this issue and thus is not a simple reuse of prior techniques. To further clarify, we provide ablated comparisons in Appendix "E.1 MORE ABLATION STUDIES", with quantitative results summerized as follows:
> | Strategy | Noise Pos. | Noise Col. | PSNR↑ | SSIM↑ | LPIPS↓ |
> |----------|-------------|------------|------|-------|--------|
> | 1        | ✗           | ✓          | **41.27** | **0.988** | **0.004** |
> | 2        | ✓           | ✓          | 40.02 | 0.985 | 0.010 |
>
> These results highlight that our 3D-consistent noise formulation is specifically designed to the point cloud rendering setting, and its design is essential for achieving high-fidelity image synthesis without the artifacts introduced by alternative approaches. Building on this formulation, our *Spatial-Aware Residual Diffusion* uses view-consistent noise as a stable initialization and extracts geometric cues from the rendered noise maps, thereby endowing the model with spatial awareness essential for high-quality point-based rendering.
>
> **3. Effectiveness of the Adaptive CoNo-Splatting module.** In addition to the 3D consistent noise, we would like to clarify that the proposed *Adaptive CoNo-Splatting* module design, though simple but very effective. In our paper, this strategy is shown to yield clear improvements, as reported in Table 5. To further evaluate its performance, we integrate it into one of our baselines, PFGS, in a plug-and-play fashion. Specifically, we remove its Gaussian regressor and the corresponding rendering pipeline, and replace them with our *Adaptive CoNo-Splatting* module. The quantitative comparison is summarized below:
> | Method              | PSNR ↑ | SSIM ↑ | LPIPS ↓ | Training (h) | Inference (fps) |
> |--------------------|-------:|-------:|--------:|-------------:|----------------:|
> | PFGS               | 19.86  | 0.758  | 0.452   | 5 + 36        | 3.6             |
> | PFGS + CoNo-Splatting | **21.05** | **0.782** | **0.433** | **2 + 17**      | **4.5**         |
>
> Notably, **both rendering quality and model efficiency improve, with the training time reducing by more than half**. This improvement stems from replacing the heavy convolutional design in Gaussian regressor with our adaptive network-free design, which simplifies the training pipeline and reduces computational overhead. In addition, our approach eliminates the training gap between the two stages in the original PFGS, leading to better overall performance.

---

> ### Author Response · Authors · 2025-11-28
> **Response to Reviewer PDkh (2/4)**
>
> Overall, we believe that our method is carefully designed to form an effective framework for point-based neural rendering, providing a new angle for solving the core challenges of the task. We hope this clarification addresses the reviewer’s concerns, and we would be glad to provide additional details if further explanation is required.
>
> [1] Min-Seop Kwak, Donghoon Ahn, Inès Hyeonsu Kim, Jin-Hwa Kim, and Seungryong Kim. *Geometry-aware score distillation via 3d consistent noising and gradient consistency modeling.* arXiv preprint arXiv:2406.16695, 2024
>
> **Comment 2.** Clarification on multi-view consistency.
>
> **Response.** We thank the reviewer for this insightful comment. The multi-view consistency of our method arises mainly from two factors. First, the view-consistent noise provides a stable initialization for the reverse process of the residual diffusion model, which starts from:
> $ I^v_{T'}=c_{T'} \cdot I^v_{c}+d_{T'} \cdot I^v_{\epsilon}. $In this formulation, both $ I^v_{c} $ and $I^v_{\epsilon}$ are view-consistent, with nearby views sharing substantial information overlap. **This correlated initialization guides the diffusion process to evolve toward similar regions of the solution space across viewpoints, leading to improved multi-view consistency**.
>
> Second, from the perspective of the network architecture, when a 3D-consistent point cloud is projected onto adjacent views, **the local receptive fields of the UNet allow it to capture and match spatially consistent structures in the 2D image plane**. As a result, local correlations across views are preserved, which improves the consistency of the predicted noise and ultimately enhances multi-view coherence in the generated images.
>
> To further address the reviewer's concern, we add a quantitative evaluation using the Thresholded Symmetric Epipolar Distance (TSED) metric [1], which quantifies the number of consistent frame pairs in a sequence. We compare with both generalizable methods and per-scene optimized methods as follows:
>
> | T  | Ours | PFGS   | NPBG++ | TriVol | 3DGS   | RPBG   |
> |---:|---------:|-------:|-------:|-------:|-------:|-------:|
> | 2  | **0.2347** | 0.2143 | 0.1837 | 0.1531 | **0.2347** | 0.2245 |
> | 4  | **0.4796** | 0.4286 | 0.3776 | 0.3265 | 0.4694 | 0.4388 |
> | 8  | **0.8776** | 0.7755 | 0.7143 | 0.6735 | 0.8571 | 0.8061 |
>
> Specifically, the TSED value under smaller $T$ primarily reflects the temporal flickering or instability between consecutive output frames, while under larger values of $T$, it corresponds to the overall consistency of the output frames across large camera view deviations. For better intuitive evaluation, we recommend that the reviewer refer to the supplementary video for clearer visualizations and comparisons.
>
> [1] Jason J. Yu, Fereshteh Forghani, Konstantinos G. Derpanis, and Marcus A. Brubaker. Long-term photometric consistent novel view synthesis with diffusion models. In *Proceedings of the IEEE/CVF International Conference on Computer Vision and Pattern Recognition*, pp. 7094–7104, 2023.
>
> **Comment 3.** Comparing the Refinement Network with Nearest-Neighbor Color Interpolation and Off-the-Shelf Image Inpainting.
>
> **Response.** We thank the reviewer for this constructive comment. Following your suggestion, we have added additional comparative experiments to more thoroughly evaluate our method. A comprehensive comparison is provided in Appendix E.2 of the revised manuscript, particularly the sub-section titled "Additional Baselines for Image-based Refinement". For clarity, we summarize the key observations here. First, due to the inherent sparse and uneven nature of points, the rendered images contain holes. Directly interpolating the rendered images leads to blury results. For images inpainter, we apply two variants. The former such as SD-XL Inpainting [1] and FLUX-ControlNet-Inpainting [2] require a mask input for missing regions guidance, which we find them fail to fill in the holes due to the densely distrubuted tiny masks, which are unseen during their training. The later, we employ a multimodal generative model (e.g., GPT-4o) to restore the rendered image using both the rendered view and text guidance as inputs. Although it can somehow produce photorealistic results, it always generates inconsistent results across different views.
>
> [1] SD-XL Inpainting, https://huggingface.co/diffusers/stable-diffusion-xl-1.0-inpainting-0.1
>
> [2] FLUX-ControlNet-Inpainting, https://github.com/alimama-creative/FLUX-Controlnet-Inpainting
>
> **Comment 4.** Clarification on Efficiency and Robustness Comparisons.
>
> **Response.** We thank the reviewer for this constructive comment. Following your suggestion, we have added additional comparative experiments to more thoroughly evaluate our method. A comprehensive comparison is provided in Table 3 of the revised manuscript. For clarity, we summarize the key quantitative results here:

---

> ### Author Response · Authors · 2025-11-28
> **Response to Reviewer PDkh (3/4)**
>
> Robustness (PSNR) with respect to point density on Thuman2.0.
> | Method  | 60k  | 80k  | 100k | 120k |
> |--------:|-----:|-----:|-----:|-----:|
> | NPBG++  | 26.34 | 27.12 | 27.33 | 27.55 |
> | TriVol  | 25.93 | 26.21 | 26.71 | 27.08 |
> | PFGS    | 36.16 | 36.45 | 36.58 | 36.59 |
> | DiffPBR-E | 39.87 | 40.96 | 41.29 | 41.72 |
> | DiffPBR-Q | **40.02** | **41.29** | **41.62** | **42.01** |
>
> Evaluation of quality (PSNR)-efficiency trade-off on the DTU (640 × 512).
> | Method     | PSNR ↑ | SSIM ↑ | LPIPS ↓ | GPU hrs ↓ | FPS ↑  |
> |-----------:|------:|-------:|--------:|----------:|-------:|
> | NPBG++     | 22.32 | 0.833  | 0.327   | ∼10       | **11**     |
> | TriVol     | 20.02 | 0.674  | 0.483   | ∼48       | 0.07   |
> | PFGS       | 25.44 | 0.901  | 0.164   | ∼40       | 0.5    |
> | DiffPBR-E  | 28.15 | 0.919  | 0.138   | **∼8**        | 9.75   |
> | DiffPBR-Q  | **28.45** | **0.935**  | **0.124**   | **∼8**        | 1.9    |
>
> **Comment 5.** Clarifying the Ill-Posed Nature of Point-Cloud Rendering and Improving the Introduction.
>
> **Response.** We thank the reviewer for this good question. As the reviewer point out, the discrete nature of point clouds makes point-based rendering an inherently ill-posed problem, which is also a fundamental limitation of purely graphics-based renderers, as stated in our introduction L73, "Third, point clouds lack explicit surface connectivity, making the rendering process highly sensitive to point-wise parameters such as scale".
>
> NeRF and 3DGS address this issue in 3D space through time-consuming per-scene optimization to obtain spatial descriptors. In contrast, our method resolves it in a feed-forward manner by dynamically learning and adjusting point scales, thereby enabling faithful splatting without scene-specific optimization. This learnable mechanism directly tackles the scale sensitivity inherent in point-based rendering and represents one of our key contributions.
>
> Following your suggestions, **we have polished our introduction** as follows: *Third, point clouds lack explicit surface connectivity, making point-cloud rendering an ill-posed problem that is highly sensitive to point-wise parameters such as scale*. We appreciate your careful review and valuable feedback, which have helped improve the quality of our manuscript.
>
> **Comment 6.** Clarification on the Role of Noise Inputs in Diffusion-Based Image Restoration.
>
> **Response.** We thank the reviewer for the careful reading and apologize for the imprecise wording in this sentence. The original statement in our paper: *“Second, diffusion models in image restoration rely on the assumption of pure noise inputs, yet degraded renderings of point clouds often retain substantial structural and color information. Reconstructing these inputs from scratch is both unnecessary and computationally inefficient, as it requires recovering fine details that are already present in the scene. This inefficiency leads to excessive computational overhead, hindering the model’s performance.”* was intended to highlight a different aspect of noise initialization rather than suggesting that existing diffusion models only take pure noise as input.
>
> Specifically, our intended meaning is as follows: Existing diffusion-based image restoration models start the reverse denoising process from independent 2D Gaussian noise, while the degraded image serves as the conditioning input. As a result, the noise initialization for different viewpoints is independent and lacks any 3D structural correlation, which prevents these models from preserving cross-view geometric consistency at the beginning of the diffusion trajectory. In contrast, **our noise image is obtained via 3D-consistent projection, ensuring that the noise patterns across different views share the same underlying geometric priors.** This correlation allows our model to better exploit the 3D structure embedded in the scene and thus enables more consistent refinement across viewpoints.
>
> We hope this clarification addresses the reviewer’s concern, and we thank the reviewer for highlighting this point. The corresponding statement has been revised accordingly in the updated manuscript.
>
> **Comment 7.** Please provide LPIPS & SSIM for the ablation and other experiments that only use PSNR – LPIPS in particular is much more reflective of perceived quality.
>
> **Response.** We thank the reviewer for this insightful and constructive comment. Following the suggestion, **we have added LPIPS and SSIM metrics to both the comparative and ablation experiments** in Appendix H of the revised manuscript. We sincerely appreciate your careful review and valuable feedback, which have helped improve the clarity and completeness of our work.

---

> ### Author Response · Authors · 2025-11-28
> **Response to Reviewer PDkh (4/4)**
>
> **Comment 8.** Many \citet or \cite should become \citep
>
> **Response.** We thank the reviewer for raising this constructive comment. Following the suggestion, we have carefully re-read the manuscript and made the corresponding revisions in the updated version. We sincerely appreciate your careful review and valuable feedback, which have helped improve the professionalism and overall quality of the paper.

---

### Author Response · Authors · 2025-12-04
**Response Summary**

Dear Area Chair and Reviewers,

We deeply appreciate your time and effort in reviewing our submission and providing thoughtful feedback and valuable insights. In particular, we are encouraged by the reviewers' recognition of our work, including:

- The *Adaptive CoNo-Splatting* is **elegant** (Reviewer PDkh), **insightful** (Reviewer Z48r), and **interesting** (Reviewer Vaw3);
- The *Spatial-Aware Residual Diffusion* is **elegant and well-motivated** (Reviewer Vaw3) and **powerful** (Reviewer 6AMT);
- The overall pipeline is **novel** (Reviewer PDkh), **well-motivated** (Reviewer 6AMT), and **well-illustrated** (Reviewer Vaw3);
- The experimental results are **satisfactory** (Reviewer Vaw3), **thorough** (Reviewer PDkh), and **self-contained and extensive** (Reviewer Z48r);
- The paper is **clear, well-structured, and pleasant to read** (Reviewer PDkh), and the pseudo-code and appendix **demonstrate commendable clarity and reproducibility** (Reviewer Vaw3).

We sincerely thank the reviewers for their constructive comments, which have greatly helped us improve this work. In the revised manuscript, we have addressed all concerns through additional experiments, theoretical analyses, and detailed discussions. Major revisions and updates are highlighted in blue for clarity. Specifically, the key responses and revisions include:

- **For novelty**, we would like to clarify that maintaining 3D consistency remains an active research direction, and many works has proposed leveraging 3D spatial information to improve rendering coherence. For example, baselines such as PFGS and NPBG++ rely on heavy feature extractors to aggregate spatial features from input point clouds or images. In contrast, we provide a **simple and "elegent" way** by introducing 3D-consistent noise to guide the residual diffusion refinement. Compared to PFGS, our design not only achieves substantially faster training (**8 hours vs. 41 hours**) but also yields higher reconstruction quality (**4~5 dB PSNR improvement**) and better multi-view consistency (**11.52% average improvement in TSED**).
Moreover, regarding the concerns about the novelty of 3D-consistent noise, neither reviewer (PDkh or 6AMT) provides any citation to substantiate this claim. We acknowledge that a concurrent work, GSD [1] apdots a similar idea in the text-to-3D generation task. However, **their problem setting is different from ours, and their strategy upsamples noise in point-cloud space, which disrupts the pixel-level alignment between the rendered RGB image and the noise field, resulting in blurry refinement artifacts and a 1.25 dB drop in PSNR** (Figure 9 and Table 10).
- **For clarity**, we have provided a clearer mathematical formulation of the CNSplat function and clarified the notation in *Spatial-Aware Residual Diffusion* (Reviewer Z48r), expanded the explanation of the two regularizers (Reviewer 6AMT), refined the abstract (Reviewer Z48r) and the introduction (Reviewer PDkh), and corrected citation formats (Reviewers PDkh, Vaw3).
- **For experiments**, we have expanded the comparative experiments by adding per-scene optimized baselines (Reviewers Z48r, 6AMT, and Vaw3), evaluations in more challenging outdoor environments (Reviewer Vaw3), and multi-view consistency metrics (Reviewers PDkh, Z48r). We further broadened the ablation studies to validate our design choices (Reviewers PDkh, Vaw3) and parameter settings (Reviewer Z48r).

We believe that our work provides an effective solution for point-based neural rendering, providing a new angle for solving the core challenges of the task. We would be happy to provide further clarifications or make additional refinements if there are any additional concerns or suggestions.

Thank you once again for your valuable time and feedback.

Kind regards,

*All the authors*

[1] Min-Seop Kwak, Donghoon Ahn, Inès Hyeonsu Kim, Jin-Hwa Kim, and Seungryong Kim. Geometry-aware score distillation via 3d consistent noising and gradient consistency modeling. arXiv preprint arXiv:2406.16695, 2024

---

> ### Author Response · Authors · 2025-12-04
> **Response Letter: Summary of Revisions and Updates (1/2)**
>
> Dear Reviewers,
>
> Thank you for your time and constructive feedback. We have revised the manuscript accordingly and re-uploaded the updated PDF. The key changes are summarized below:
>
> - **For Reviewer PDkh:**
>   - We conducted a plug-and-play experiment demonstrating that *Adaptive CoNo-Splatting* improves training efficiency (**from 41 to 19 hours**), inference speed (**from 3.6 to 4.5 FPS**), and rendering quality (**from 19.86 to 21.05 PSNR**) when integrated into PFGS (our most related baseline), as shown in Appendix E.1 (Table 13 and Figure 11).
>   - We evaluated **multi-view consistency** using the Thresholded Symmetric Epipolar Distance (TSED), showing that **our method outperforms generalizable methods** (e.g., PFGS, **with an average 11.52% TSED improvement**) and is **comparable to or even better than per-scene optimized methods** (e.g., 3DGS, **with an average 2.28% TSED improvement**) (Appendix E.2, Table 18).
>   - We demonstrated **better rendering quality and visual consistency** compared to **nearest-neighbor interpolation** and three **inpainting pipelines** (two mask-guided variants based on Stable Diffusion XL and FLUX, and one multimodal generative model, GPT-4o) in Appendix E.1, Figure 10.
>   - We evaluated **efficiency and robustness with respect to point-cloud density**, showing that our method **outperforms all baselines in rendering quality** (e.g., PFGS, **with an average 4.79 dB PSNR improvement**) while achieving **efficiency comparable to the best-performing baseline** (NPBG++: 11 FPS vs. ours: 9.75 FPS) (Appendix E.2, Table 15; Section 4.2, Table 3).
>   - We reported **full PSNR, SSIM, and LPIPS metrics** for all ablations and other experiments, ensuring a more comprehensive evaluation of our method (Appendix H, Tables 20~23).
>   - We refined the introduction to more clearly articulate the **ill-posed nature** of rendering sparse point clouds and corrected the misused citation formats.
>
> - **For Reviewer Z48r:**
>   - We conducted a **bilateral comparison on dense vs. sparse point clouds**, showing that our generally trained model can **outperform per-scene optimized methods** (e.g., 3DGS, **with an average 0.81 dB PSNR improvement**) (Appendix E.2, Table 17).
>   - We conducted a **qualitative comparison with the graphics-based pipeline** (Mitsuba), demonstrating our method's **superior rendering quality** and highlighting the necessity of the diffusion-based refinement stage (Appendix E.2, Figure 12).
>   - We conducted a more thorough **ablation study on heuristic truncation and hyper-parameter choices** for *Adaptive CoNo-Splatting*, further **validating the effectiveness of our design choices and parameter settings** (Appendix E.1, Tables 11 to 12).
>   - We evaluated **multi-view consistency** using the Thresholded Symmetric Epipolar Distance (TSED), showing that our method **outperforms generalizable methods** (e.g., PFGS, **with an average 11.52% TSED improvement**) and is **comparable to or even better than per-scene optimized methods** (e.g., 3DGS, with an average 2.28% TSED improvement) (Appendix E.2, Table 18).
>   - We provided a **clearer mathematical formulation** of the CNSplat function in Equation 2 (Section 3.1) and more **complete symbol definitions** in Appendix A.
>   - We refined the **efficiency statement** in the abstract to improve clarity.
>
> - **For Reviewer 6AMT:**
>   - We conducted additional comparative experiments with per-scene optimized baselines (3DGS and RPBG), showing that our method **outperforms them on challenging datasets** with poor image quality (ScanNet, **~1 dB PSNR improvement**),  incomplete (DTU, **2~3 dB PSNR improvement**) or unbounded (Tanks and Temples, **1~2 dB PSNR improvement**) point clouds. Meanwhile, on the well-conditioned dataset Thuman2.0, our method is **comparable to 3DGS and 4~5 dB PSNR improvement over RPBG** (Appendix E.2, Tables 16 & 19; Figures 13 & 14).
>   - We conducted a plug-and-play experiment demonstrating that *Adaptive CoNo-Splatting* improves training efficiency (**from 41 to 19 hours**), inference speed (**from 3.6 to 4.5 FPS**), and rendering quality (**from 19.86 to 21.05 PSNR**) when integrated into PFGS (our most related baseline), as shown in Appendix E.1 (Table 13 and Figure 11).
>   - We revised the explanation of the two scale regularizers, replacing the previous notion of an *adversarial balance* with *complementary interplay* (Section 3.1).

---

> > ### Author Response · Authors · 2025-12-04
> > **Response Letter: Summary of Revisions and Updates (2/2)**
> >
> > - **For Reviewer Vaw3:**
> >   - We conducted additional comparative experiments with per-scene optimized baselines (3DGS and RPBG), showing that our method **outperforms them on challenging datasets** with poor image quality (ScanNet, **~1 dB PSNR improvement**), incomplete (DTU, **2~3 dB PSNR improvement**) or unbounded (Tanks and Temples, **1~2 dB PSNR improvement**) point clouds. Meanwhile, on the well-conditioned dataset Thuman2.0, our method is **comparable to 3DGS and 4~5 dB PSNR improvement over RPBG** (Appendix E.2, Tables 16 & 19; Figures 13 & 14).
> >   - We conducted a plug-and-play experiment demonstrating that *Adaptive CoNo-Splatting* improves training efficiency (**from 41 to 19 hours**), inference speed (**from 3.6 to 4.5 FPS**), and rendering quality (**from 19.86 to 21.05 PSNR**) when integrated into PFGS (our most related baseline), as shown in Appendix E.1 (Table 13 and Figure 11).
> >   - We corrected previously misused citation formats and added a limitation statement regarding the current rendering FPS in Appendix G.
> >
> > We believe that the above revisions, together with the detailed responses provided in the rebuttal, comprehensively address all concerns raised by the reviewers. We sincerely appreciate the reviewers' constructive comments, which have helped us further strengthen the clarity, rigor, and completeness of the manuscript. Thank you again for your thoughtful feedback and for the opportunity to improve our work.

---

### Meta-Review · Area_Chair_wD27 · 2026-01-10

**Summary:**

This paper presents DiffPBR, a diffusion-based framework that synthesizes coherent, photorealistic renderings from diverse point-cloud inputs. Two reviewers are positive, while the other two are slightly negative. The authors provided detailed and strong responses to all comments, and the technical concerns could be fully resolved. Depending on reviewer preferences, the novelty may be regarded as incremental. Considering the overall scores and the rebuttal, AC believes this paper is above the acceptance bar and should be accepted as a poster.

**Reviewer Concerns:**

Reviewer PDkh (marginally below acceptance) raised some concerns, for example, the conceptual contribution is limited, multi-view consistency is encouraged but not guaranteed, and key baselines/ablations were missing. The authors responded strongly with new baselines, broader efficiency/robustness tables, and an explicit ill-posedness discussion, plus a quantitative multi-view metric (TSED). These address most empirical/completeness issues.

Reviewer Z48r (marginally above acceptance) is positive, yet has concerns about the training/generalization premise, missing comparisons to strong NVS baselines (3DGS/FSGS/RPBG), insufficient multi-view consistency quantification. The authors added TSED, included per-scene baselines (3DGS/RPBG) and Tanks & Temples experiments, ran robustness tests to registration noise and percentile/mean scale ablations. These can resolve most technical and experimental gaps.

Reviewer 6AMT (marginally below acceptance) has concerns on limited novelty (residual learning + 3D consistency), missing SOTA comparisons (3DGS/variants), potentially misleading efficiency claims, and unclear/stability questions around joint optimization of splatting parameters with diffusion. The authors added direct 3DGS/RPBG comparisons across datasets, clarified efficiency via a plug-and-play PFGS+CoNo experiment, and provided ablations showing gains from adaptive scale regulation and regularizers vs. fixed heuristics. These address the actionable technical points well.

Reviewer Vaw3 (accept) is quite positive. The concerns were about weak coupling between the two contributions, insufficient benchmark diversity and missing 3DGS baseline. The authors directly addressed all of these with added datasets/baselines (including Tanks & Temples, 3DGS/RPBG) and a clearer coupling argument (probabilistic splatting producing the right conditioning/noise for residual diffusion). These concerns appear fully resolved.

**Reviewer Scores:**

Given the strong and detailed rebuttal, Reviewer PDkh and Reviewer 6AMT are likely to raise the score to Marginally above acceptance bar, or keep the score unchanged if they weigh more on the novelty. Reviewer Z48r and Reviewer Vaw3 are very likely to keep the score unchanged.

---

### Decision · Program_Chairs · 2026-01-26

Accept (Poster)